# Convex Optimization Procedure for Clustering: Theoretical Revisit

**Changbo Zhu**
Department of Electrical and Computer Engineering
Department of Mathematics
National University of Singapore
elezhuc@nus.edu.sg

**Huan Xu**
Department of Mechanical Engineering
National University of Singapore
mpexuh@nus.edu.sg

**Chenlei Leng**
Department of Statistics
University of Warwick
c.leng@warwick.ac.uk

**Shuicheng Yan**
Department of Electrical and Computer Engineering
National University of Singapore
eleyans@nus.edu.sg

## Abstract

In this paper, we present theoretical analysis of SON – a convex optimization procedure for clustering using a sum-of-norms (SON) regularization recently proposed in [8, 10, 11, 17]. In particular, we show if the samples are drawn from two cubes, each being one cluster, then SON can provably identify the cluster membership provided that the distance between the two cubes is larger than a threshold which (linearly) depends on the size of the cube and the ratio of numbers of samples in each cluster. To the best of our knowledge, this paper is the first to provide a rigorous analysis to understand why and when SON works. We believe this may provide important insights to develop novel convex optimization based algorithms for clustering.

## 1 Introduction

Clustering is an important problem in unsupervised learning that deals with grouping observations (data points) appropriately based on their similarities or distances [20]. Many clustering algorithms have been proposed in literature, including K-means, spectral clustering, Gaussian mixture models and hierarchical clustering, to solve problems with respect to a wide range of cluster shapes. However, much research has pointed out that these methods all suffer from instabilities [3, 20, 16, 15, 13, 19]. Taking K-means as an example, the formulation of K-means is NP-hard and the typical way to solve it is the Lloyd's method, which requires randomly initializing the clusters. However, different initialization may lead to significantly different final cluster results.

### 1.1 A Convex Optimization Procedure for Clustering

Recently, Lindsten et al. [10, 11], Hocking et al. [8] and Pelckmans et al. [17] proposed the following convex optimization procedure for clustering, which is termed as SON by Lindsten et al. [11] (Also called Clusterpath by Hocking et al. [8]),

$$\hat{\mathbf{X}} = \arg \min_{\mathbf{X} \in \mathbb{R}^{n \times p}} \|\mathbf{A} - \mathbf{X}\|_F^2 + \alpha \sum_{i<j} \|\mathbf{X}_{i\cdot} - \mathbf{X}_{j\cdot}\|_2. \tag{1}$$

Here $\mathbf{A}$ is a given data matrix of dimension $n \times p$ where each row is a data point, $\alpha$ is a tunable parameter to determine the number of clusters, $\|\cdot\|_F$ denotes the Frobenius norm and $\mathbf{X}_{i\cdot}$ denotes the $i$th row of $\mathbf{X}$.

The main idea of the algorithm is that if the $i$-th sample and the $j$-th sample belong to the same cluster, then $\hat{\mathbf{X}}_{i\cdot}$ and $\hat{\mathbf{X}}_{j\cdot}$ should be equal. Intuitively, this is due to the fact that the second term is a regularization term that enforces the rows of $\hat{\mathbf{X}}$ to be the same, and can be seen as a generalization of the fused Lasso penalty [18]. In particular, this penalty seeks to fuse the rows of $\hat{\mathbf{X}}$. From another point of view, the regularization term can be seen as an $\ell_{1,2}$ norm, i.e., the sum of $\ell_2$ norm. Such a norm is known to encourage block sparse (in this case row-sparse) solutions [1]. Thus, it is expected that for many $(i,j)$ pairs, $\hat{\mathbf{X}}_{i\cdot} - \hat{\mathbf{X}}_{j\cdot} = \mathbf{0}$.

Mathematically, given $c$ disjoint clusters $\{\mathbb{C}^1, \mathbb{C}^2, \cdots, \mathbb{C}^c\}$ with $\mathbb{C}^i \subseteq \mathbb{R}^p$ for $i = 1, 2, \cdots, c$, we define the Cluster Membership Matrix of a given data matrix $\mathbf{A}$ as the following.

**Definition 1.** *Given a data matrix* $\mathbf{A}$ *of dimension* $n \times p$, *for* $j = 1, 2, \cdots, c$, *set* $\mathbb{I}_j = \{i \mid \mathbf{A}_{i\cdot} \in \mathbb{C}^j, 1 \leq i \leq n\}$. *We say that a matrix* $\mathbf{X}$ *of dimension* $n \times p$ *is a **Cluster Membership Matrix** of* $\mathbf{A}$ *if*

$$\begin{cases} \mathbf{X}_{i\cdot} = \mathbf{X}_{j\cdot} & \text{if } i \in \mathbb{I}_k, \ j \in \mathbb{I}_k \text{ and } 1 \leq k \leq c \\ \mathbf{X}_{i\cdot} \neq \mathbf{X}_{j\cdot} & \text{if } i \in \mathbb{I}_m, j \in \mathbb{I}_l, 1 \leq m \leq c, 1 \leq l \leq c \text{ and } m \neq l. \end{cases}$$

Given a data matrix $\mathbf{A}$, if the optimal solution $\hat{\mathbf{X}}$ of Problem (1) is a Cluster Membership Matrix of $\mathbf{A}$, then we can determine the cluster membership by simply grouping the identical rows of $\hat{\mathbf{X}}$ together. We say that SON successfully recovers the cluster membership of $\mathbf{A}$ in this case.

Notice that unlike previous approaches, SON does not suffer from the instability issue since it is a strictly convex optimization problem and the solution is fixed once a data matrix $\mathbf{A}$ is given. Moreover, SON can easily be adapted to incorporate a priori knowledge of the clustering membership. For example, if we have prior knowledge about which points are more likely to be in the same cluster, we can appropriately weight the regularization term, i.e., change the regularization term to $\alpha \sum_{i<j} \gamma_{ij} \|\mathbf{X}_{i\cdot} - \mathbf{X}_{j\cdot}\|_2$ for some $\gamma_{ij} > 0$.

The main contribution of this paper is to provide theoretic analysis of SON, in particular to derive sufficient conditions when SON successfully recovers the clustering membership. We show that if there are two clusters, each of which is a cube, then SON succeeds provided that the distance between the cubes is larger than a threshold value that depends on the cube size and the ratio of number of samples drawn in each cluster. Thus, the intuitive argument about why SON works is made rigorous and mathematically solid. To the best of our knowledge, this is the first attempt to theoretically quantify why and when SON succeeds.

**Related Work:** we briefly review the related works on SON. Hocking et al. [8] proposed SON, arguing that it can be seen as a generalization of hierarchical clustering, and presented via numerical simulations several situations in which SON works while K-means and average linkage hierarchical clustering fail. They also developed R package called "clusterpath" which can be used to solve Problem (1). Independently, Lindsten et al. [10, 11] derived SON as a convex relaxation of K-means clustering. In the algorithmic aspect, Chi et al. [6] developed two methods to solve Problem (1), namely, Alternating Direction Method of Multipliers (ADMM) and alternating minimization algorithm (AMA). Marchetti et al. [14] generalized SON to the high-dimensional and noisy cases. Yet, in all these works, no attempt has been made to study rigorously why and when SON succeeds.

**Notation:** in this paper, matrices are denoted by upper case boldface letters (e.g. $\mathbf{A}$, $\mathbf{B}$), sets are denoted by blackboard bold characters (e.g. $\mathbb{R}$, $\mathbb{I}$, $\mathbb{C}$) and operators are denoted by Fraktur characters (e.g. $\mathfrak{D}$, $\mathfrak{M}$). Given a matrix $\mathbf{A}$, we use $\mathbf{A}_{i\cdot}$ to denote its $i$th row, and $\mathbf{A}_{\cdot j}$ to denote its $j$th column. Its $(i, j)$th entry is denoted by $\mathbf{A}_{i,j}$. Two norms are used: we use $\|\cdot\|_F$ to denote the Frobenius norm and $\|\cdot\|_2$ to denote the $l_2$ norm of a vector. The space spanned by the rows of $\mathbf{A}$ is denoted by Row($\mathbf{A}$). Moreover, given a matrix $\mathbf{A}$ of dimension $n \times p$ and a function $f : \mathbb{R}^p \mapsto \mathbb{R}^q$, we use the notation $f(\mathbf{A})$ to denote the matrix whose $i$th row is $f(\mathbf{A}_{i\cdot})$.

## 2   Main Result

In this section we present our main theoretic result – a provable guarantee when SON succeeds in identifying cluster membership.

## 2.1 Preliminaries

We first define some operators that will be frequently used in the remainder of the paper.

**Definition 2.** *Given any two matrices* $\mathbf{E}$ *of dimension* $n_1 \times p$ *and* $\mathbf{F}$ *of dimension* $n_2 \times p$*, define the **difference operator** $\mathfrak{D}_1$ on* $\mathbf{E}$*,* $\mathfrak{D}_2$ *on the two matrices* $\mathbf{E}, \mathbf{F}$ *and* $\mathfrak{D}$ *on the matrix constructed by concatenating* $\mathbf{E}$ *and* $\mathbf{F}$ *vertically as*

$$\mathfrak{D}_1(\mathbf{E}) = \begin{pmatrix} \mathbf{E}_{1\cdot} - \mathbf{E}_{2\cdot} \\ \mathbf{E}_{1\cdot} - \mathbf{E}_{3\cdot} \\ \vdots \\ \mathbf{E}_{1\cdot} - \mathbf{E}_{n_1\cdot} \\ \mathbf{E}_{2\cdot} - \mathbf{E}_{3\cdot} \\ \vdots \\ \mathbf{E}_{2\cdot} - \mathbf{E}_{n_1\cdot} \\ \vdots \\ \mathbf{E}_{(n_1-1)\cdot} - \mathbf{E}_{n_1\cdot} \end{pmatrix}, \mathfrak{D}_2(\mathbf{E}, \mathbf{F}) = \begin{pmatrix} \mathbf{E}_{1\cdot} - \mathbf{F}_{1\cdot} \\ \mathbf{E}_{1\cdot} - \mathbf{F}_{2\cdot} \\ \vdots \\ \mathbf{E}_{1\cdot} - \mathbf{F}_{n_2\cdot} \\ \mathbf{E}_{2\cdot} - \mathbf{F}_{1\cdot} \\ \vdots \\ \mathbf{E}_{2\cdot} - \mathbf{F}_{n_2\cdot} \\ \vdots \\ \mathbf{E}_{n_1\cdot} - \mathbf{F}_{n_2\cdot} \end{pmatrix} \text{ and } \mathfrak{D}(\begin{pmatrix} \mathbf{E} \\ \mathbf{F} \end{pmatrix}) = \begin{pmatrix} \mathfrak{D}_1(\mathbf{E}) \\ \mathfrak{D}_1(\mathbf{F}) \\ \mathfrak{D}_2(\mathbf{E}, \mathbf{F}) \end{pmatrix}.$$

In words, the operator $\mathfrak{D}_1$ calculates the difference between every two rows of a matrix and lists the results in the order indicated in the definition. Similarly, given two matrices $\mathbf{E}$ and $\mathbf{F}$, the operator $\mathfrak{D}_2(\mathbf{E}, \mathbf{F})$ calculates the difference of any two rows between $\mathbf{E}$ and $\mathbf{F}$, one from $\mathbf{E}$ and the other from $\mathbf{F}$. We also define the following average operation which calculates the mean of the row vectors.

**Definition 3.** *Given any matrix* $\mathbf{E}$ *of dimension* $n \times p$*, define the **average operator** on* $\mathbf{E}$ *as*

$$\mathfrak{M}(\mathbf{E}) = \frac{1}{n}(\sum_{i=1}^{n} \mathbf{E}_{i\cdot}).$$

**Definition 4.** *A matrix* $\mathbf{E}$ *is called **column centered** if* $\mathfrak{M}(\mathbf{E}) = \mathbf{0}$*.*

## 2.2 Theoretical Guarantees

Our main result essentially says that when there are two clusters, each of which is a cube, and they are reasonably separated away from each other, then SON successfully recovers the cluster membership. We now make this formal. For $i = 1, 2$, suppose $\mathbb{C}^i \subseteq \mathbb{R}^p$ is a cube with center $(\mu_{i1}, \mu_{i2}, \cdots, \mu_{ip})$ and edge length $\mathbf{s}_i = 2(\sigma_{i1}, \sigma_{i2}, \cdots, \sigma_{ip})$, i.e.,

$$\mathbb{C}^i = [\mu_{i1} - \sigma_{i1}, \mu_{i1} + \sigma_{i1}] \times \cdots \times [\mu_{ip} - \sigma_{ip}, \mu_{ip} + \sigma_{ip}].$$

**Definition 5.** *The **distance** $d_{1,2}$ between cubes $\mathbb{C}^1$ and $\mathbb{C}^2$ is*

$$d_{1,2} \triangleq \inf\{\|\mathbf{x} - \mathbf{y}\|_2 \mid \mathbf{x} \in \mathbb{C}^1, \mathbf{y} \in \mathbb{C}^2\}.$$

**Definition 6.** *The **weighted size** $w_{1,2}$ with respect to $\mathbb{C}^1$, $\mathbb{C}^2$, $n_1$ and $n_2$ is defined as*

$$w_{1,2} = \max\left\{\left(\frac{2n_2(n_1 - 1)}{n_1^2} + 1\right)\|\mathbf{s}_1\|_2, \left(\frac{2n_1(n_2 - 1)}{n_2^2} + 1\right)\|\mathbf{s}_2\|_2\right\}.$$

**Theorem 1.** *Given a column centered data matrix* $\mathbf{A}$ *of dimension* $n \times p$*, where each row is arbitrarily picked from either cube* $\mathbb{C}^1$ *or cube* $\mathbb{C}^2$ *and there are totally* $n_i$ *rows chosen from* $\mathbb{C}^i$ *for* $i = 1, 2$*, if* $w_{1,2} < d_{1,2}$*, then by choosing the parameter* $\alpha \in \mathbb{R}$ *such that* $w_{1,2} < \frac{n}{2}\alpha < d_{1,2}$*, we have the following:*

1. *SON can correctly determine the cluster membership of* $\mathbf{A}$*;*

2. *Rearrange the rows of* $\mathbf{A}$ *such that*

$$\mathbf{A} = \begin{pmatrix} \mathbf{A}^1 \\ \mathbf{A}^2 \end{pmatrix} \text{ and } \mathbf{A}^i = \begin{pmatrix} \mathbf{A}^i_{1\cdot} \\ \mathbf{A}^i_{2\cdot} \\ \vdots \\ \mathbf{A}^i_{n_i\cdot} \end{pmatrix}, \tag{2}$$

*where for $i = 1, 2$ and $j = 1, 2, \cdots, n_i$, $\mathbf{A}^i_{j\cdot} = (\mathbf{A}^i_{j,1}, \mathbf{A}^i_{j,2}, \cdots, \mathbf{A}^i_{j,p}) \in \mathbb{C}^i$. Then, the optimal solution $\hat{\mathbf{X}}$ of Problem (1) is given by*

$$\hat{\mathbf{X}}_{i\cdot} = \begin{cases} \frac{n_2}{n_1+n_2}\left(1 - \frac{n\alpha}{2\|\mathfrak{M}(\mathfrak{D}_2(\mathbf{A}^1,\mathbf{A}^2))\|_2}\right)\mathfrak{M}\left(\mathfrak{D}_2(\mathbf{A}^1,\mathbf{A}^2)\right), & \text{if } \mathbf{A}_{i\cdot} \in \mathbb{C}^1; \\ -\frac{n_1}{n_1+n_2}\left(1 - \frac{n\alpha}{2\|\mathfrak{M}(\mathfrak{D}_2(\mathbf{A}^1,\mathbf{A}^2))\|_2}\right)\mathfrak{M}\left(\mathfrak{D}_2(\mathbf{A}^1,\mathbf{A}^2)\right), & \text{if } \mathbf{A}_{i\cdot} \in \mathbb{C}^2. \end{cases}$$

The theorem essentially states that we need $d_{1,2}$ to be large and $w_{1,2}$ to be small for correct determination of the cluster membership of $\mathbf{A}$. This is indeed intuitive. Notice that $d_{1,2}$ is the distance between the cubes and $w_{1,2}$ is a constant that depends on the size of the cube as well as the ratio between the samples in each cube. Obviously, if the cubes are too close with each other, i.e., $d_{1,2}$ is small, or if the sizes of the clusters are too big compared to their distance, it is difficult to determine the cluster membership correctly. Moreover, when $n_1 \ll n_2$ or $n_1 \gg n_2$, $w_{1,2}$ is large, and the theorem states that it is difficult to determine the cluster membership. This is also well expected, since in this case one cluster will be overwhelmed by the other, and hence determining where the data points are chosen from becomes problematic.

The assumption in Theorem 1 that the data matrix $\mathbf{A}$ is column centered can be easily relaxed, using the following proposition which states that the result of SON is invariant to any isometry operation.

**Definition 7.** *An **isometry** of $\mathbb{R}^n$ is a function $f : \mathbb{R}^n \to \mathbb{R}^n$ that preserves the distance between vectors, i.e.,*

$$\|f(\mathbf{u}) - f(\mathbf{w})\|_2 = \|\mathbf{u} - \mathbf{w}\|_2, \forall\, \mathbf{u}, \mathbf{w} \in \mathbb{R}^n.$$

**Proposition 1. (Isometry Invariance)** *Given a data matrix $\mathbf{A}$ of dimension $n \times p$ where each row is chosen from some cluster $\mathbb{C}^i, i = 1, 2, \cdots, c$, and $f(\cdot)$ an isometry of $\mathbb{R}^p$, we have*

$$\hat{\mathbf{X}} = \arg\min_{\mathbf{X}\in\mathbb{R}^{n\times p}} \|\mathbf{A} - \mathbf{X}\|_F^2 + \alpha \sum_{i<j} \|\mathbf{X}_{i\cdot} - \mathbf{X}_{j\cdot}\|_2$$

$$\Longleftrightarrow f(\hat{\mathbf{X}}) = \arg\min_{\mathbf{X}\in\mathbb{R}^{n\times p}} \|f(\mathbf{A}) - \mathbf{X}\|_F^2 + \alpha \sum_{i<j} \|\mathbf{X}_{i\cdot} - \mathbf{X}_{j\cdot}\|_2.$$

*This further implies that if SON successfully determines the cluster membership of $\mathbf{A}$, then it also successfully determines the cluster membership of $f(\mathbf{A})$.*

## 3 Kernelization

SON can be easily kernelized as we show in this section. In the kernel clustering setup, instead of clustering $\{\mathbf{A}_{i\cdot}\}$ such that points within a cluster are closer *in the original space*, we want to cluster $\{\mathbf{A}_{i\cdot}\}$ such that points within a cluster are closer *in the feature space*. Mathematically, this means we map $\mathbf{A}_{i\cdot}$ to a Hilbert space $\mathcal{H}$ (the feature space) by the feature mapping function $\phi(\cdot)$ and perform clustering on $\{\phi(\mathbf{A}_{i\cdot})\}$.

Notice that we can write Problem (1) in terms of the inner product $\langle \mathbf{A}_{i\cdot}, \mathbf{A}_{j\cdot} \rangle$, $\langle \mathbf{A}_{i\cdot}, \mathbf{X}_{j\cdot} \rangle$ and $\langle \mathbf{X}_{i\cdot}, \mathbf{X}_{j\cdot} \rangle$. Thus, for SON in the feature space, we only need to replace all these inner products by $\langle \phi(\mathbf{A}_{i\cdot}), \phi(\mathbf{A}_{j\cdot}) \rangle$, $\langle \phi(\mathbf{A}_{i\cdot}), \mathbf{X}_{j\cdot} \rangle$ and $\langle \mathbf{X}_{i\cdot}, \mathbf{X}_{j\cdot} \rangle$. Thus, SON in the feature space can be formulated as

$$\hat{\mathbf{X}} = \arg\min_{\mathbf{X}\in\mathbb{R}^{n\times q}} \sum_{i=1}^{n} \left(\langle \phi(\mathbf{A}_{i\cdot}), \phi(\mathbf{A}_{i\cdot}) \rangle - 2\langle \phi(\mathbf{A}_{i\cdot}), \mathbf{X}_{i\cdot} \rangle + \langle \mathbf{X}_{i\cdot}, \mathbf{X}_{i\cdot} \rangle\right)$$
$$+ \alpha \sum_{i<j} \sqrt{\langle \mathbf{X}_{i\cdot}, \mathbf{X}_{i\cdot} \rangle - 2\langle \mathbf{X}_{i\cdot}, \mathbf{X}_{j\cdot} \rangle + \langle \mathbf{X}_{j\cdot}, \mathbf{X}_{j\cdot} \rangle}. \tag{3}$$

We have the following representation theorem about the optimal solution of (3).

**Theorem 2. (Representation Theorem)** *Each row of the optimal solution of Problem (3) can be written as a linear combination of rows of $\mathbf{A}$, i.e.,*

$$\hat{\mathbf{X}}_{i\cdot} = \sum_{j=1}^{n} a_{ij}\phi(\mathbf{A}_{j\cdot}).$$

Thus, to solve SON in the feature space reduces to finding the optimal weight $\{a_{ij}\}$. Define the kernel function as $\mathfrak{K}(\mathbf{x}, \mathbf{y}) = \langle \phi(\mathbf{x}), \phi(\mathbf{y}) \rangle$. Then Problem (3) is equivalent to

$$
\min_{\{a_{ij}\}} \sum_{i=1}^{n} \left( \mathfrak{K}(\mathbf{A}_{i \cdot}, \mathbf{A}_{i \cdot}) - 2 \sum_{k=1}^{n} a_{ik} \mathfrak{K}(\mathbf{A}_{i \cdot}, \mathbf{A}_{k \cdot}) + \sum_{k=1}^{n} \sum_{l=1}^{n} a_{ik} a_{il} \mathfrak{K}(\mathbf{A}_{k \cdot}, \mathbf{A}_{l \cdot}) \right) \\
+ \alpha \sum_{i<j} \sqrt{\sum_{k=1}^{n} \sum_{l=1}^{n} \mathfrak{K}(\mathbf{A}_{k \cdot}, \mathbf{A}_{l \cdot})(a_{ik} a_{il} - 2 a_{ik} a_{jl} + a_{jk} a_{jl})},
\tag{4}
$$

which is a second order cone program since the kernel is positive semi-definite. Notice that this implies that solving SON in the feature space only requires knowing the kernel function rather than the feature mapping $\phi(\cdot)$.

# 4 Proof

We sketch the proof of Theorem 1 here. The detailed proof is given in the supplementary material.

## 4.1 Preliminaries

We first introduce some notations useful in the proof. We use $\mathbf{I}_n$ to denote an identity matrix of dimension $n \times n$ and use $\mathbf{1}_{m \times n}$ to denote a matrix of dimension $m \times n$ with all entries being 1. Similarly, we use $\mathbf{0}_{m \times n}$ to denote a matrix of dimension $m \times n$ with all entries being 0.

We now define some special matrices. Let $\mathbf{H}_n$ denote a matrix of dimension $(n-1) \times n$ which is constructed by concatenating $\mathbf{1}_{(n-1) \times 1}$ and $-\mathbf{I}_{n-1}$ horizontally, i.e., $\mathbf{H}_n = (\mathbf{1}_{(n-1) \times 1} \quad -\mathbf{I}_{n-1})$. For $i = 1, 2, \cdots, n-2$, we first concatenate matrices $\mathbf{H}_{n-i}$ and $\mathbf{0}_{(n-1-i) \times i}$ horizontally to form a matrix $(\mathbf{0}_{(n-1-i) \times i} \quad \mathbf{H}_{n-i})$. Then, we construct $\mathbf{R}_n$ by concatenating $\{\mathbf{H}_n, (\mathbf{0}_{(n-2) \times 1} \quad \mathbf{H}_{n-1}), \cdots, (\mathbf{0}_{1 \times (n-2)} \quad \mathbf{H}_2)\}$ vertically, i.e.,

$$
\mathbf{R}_n \triangleq \begin{pmatrix} \mathbf{H}_n \\ \mathbf{0}_{(n-2) \times 1} \ \mathbf{H}_{n-1} \\ \mathbf{0}_{(n-3) \times 2} \ \mathbf{H}_{n-2} \\ \vdots \\ \mathbf{0}_{1 \times (n-2)} \ \mathbf{H}_2 \end{pmatrix}.
$$

We concatenate $m$ copies of $-\mathbf{I}_n$ vertically to form a new matrix and denote it by $\mathbf{W}_{mn \times n}$. Let $\mathbf{G}_{m,n,i}$ denote an $m \times n$ dimensional matrix where the entries of the $i$th column all equal 1 and all the other entries equal 0, i.e., $\mathbf{G}_{m,n,i} \triangleq (\mathbf{0}_{m \times (i-1)} \ \mathbf{1}_{m \times 1} \ \mathbf{0}_{m \times (n-i)})$. Then, we concatenate $\{\mathbf{G}_{m,n,1}, \mathbf{G}_{m,n,2}, \cdots, \mathbf{G}_{m,n,n}\}$ vertically and denote it by $\mathbf{S}_{mn \times n}$, i.e.,

$$
\mathbf{W}_{mn \times n} \triangleq \begin{pmatrix} -\mathbf{I}_n \\ -\mathbf{I}_n \\ \vdots \\ -\mathbf{I}_n \end{pmatrix}, \quad \mathbf{S}_{mn \times n} \triangleq \begin{pmatrix} \mathbf{G}_{m,n,1} \\ \mathbf{G}_{m,n,2} \\ \vdots \\ \mathbf{G}_{m,n,n} \end{pmatrix}.
$$

Finally, set $\mathbf{\Omega} \triangleq$

$$
\begin{pmatrix}
\mathbf{R}_{n_1-1} & \mathbf{I}_{\binom{n_1-1}{2}} & \mathbf{0}_{\binom{n_1-1}{2} \times \binom{n_2}{2}} & \mathbf{0}_{\binom{n_1-1}{2} \times n_2} & \mathbf{0}_{\binom{n_1-1}{2} \times (n_1-1)n_2} \\
\mathbf{0}_{\binom{n_2}{2} \times (n_1-1)} & \mathbf{0}_{\binom{n_2}{2} \times \binom{n_1-1}{2}} & \mathbf{I}_{\binom{n_2}{2}} & \mathbf{R}_{n_2} & \mathbf{0}_{\binom{n_2}{2} \times (n_1-1)n_2} \\
\mathbf{S}_{(n_1-1)n_2 \times (n_1-1)} & \mathbf{0}_{(n_1-1)n_2 \times \binom{n_1-1}{2}} & \mathbf{0}_{(n_1-1)n_2 \times \binom{n_2}{2}} & \mathbf{W}_{(n_1-1)n_2 \times n_2} & \mathbf{I}_{(n_1-1)n_2}
\end{pmatrix}.
$$

## 4.2 Proof sketch of Theorem 1

The proof of Theorem 1 is based on the idea of "lifting". That is, we project Problem (1) into a higher dimensional space (in particular, from $n$ rows to $n(n-1)/2$ rows), which then allows us to separate the regularization term into the sum of $l_2$ norm of each row. Although this brings additional

linear constraints to the formulation, it facilitates the analysis. In the following, we divide the proof into 3 steps and explain the main idea of each step.

**Step 1:** In this step, we derive an equivalent form of Problem (1) and give optimality conditions. For convenience, set $\mathbf{B}^{(1,2)} = \mathfrak{D}_2(\mathbf{A}^1, \mathbf{A}^2)$, $\mathbf{B}^1 = \mathfrak{D}_1(\mathbf{A}^1)$, $\mathbf{B}^2 = \mathfrak{D}_1(\mathbf{A}^2)$ and $\mathbb{V} = \{\mathbf{y} \in \mathbb{R}^{\binom{n}{2}} \mid \mathbf{\Omega}\mathbf{y} = 0\}$. The following lemmas show that we can lift the original problem into an equivalent problem that is easier to analyze.

**Lemma 1.** *If the data matrix* $\mathbf{A}$ *is column centered, then the optimal solution* $\hat{\mathbf{X}}$ *of problem* (1) *is also column centered. Further more, set* $\mathbf{B} = \mathfrak{D}(\mathbf{A})$ *and* $\hat{\mathbf{Y}} = \mathfrak{D}(\hat{\mathbf{X}})$*, we have*

$$\|\mathbf{A} - \hat{\mathbf{X}}\|_F^2 = \sum_{i=1}^{\frac{n(n-1)}{2}} \frac{1}{n}\|\mathbf{B}_{i\cdot} - \hat{\mathbf{Y}}_{i\cdot}\|_2^2.$$

**Lemma 2.** *Given a column centered data matrix* $\mathbf{A}$*, set* $\mathbf{B} = \mathfrak{D}(\mathbf{A})$ *and* $\mathbb{S} = \{\mathbf{Z} \in \mathbb{R}^{\binom{n}{2} \times p} \mid \mathbf{\Omega}\mathbf{Z}_{\cdot j} = 0, \ 1 \le j \le p\}$*. Then,* $\hat{\mathbf{X}}$ *is the optimal solution to Problem* (1) *iff*

$$\mathfrak{D}(\hat{\mathbf{X}}) = \arg\min_{\mathbf{Y} \in \mathbb{S}} \sum_{i=1}^{\frac{n(n-1)}{2}} (\frac{1}{n}\|\mathbf{B}_{i\cdot} - \mathbf{Y}_{i\cdot}\|_2^2 + \alpha\|\mathbf{Y}_{i\cdot}\|_2). \tag{5}$$

Thus, we can determine whether $\hat{\mathbf{X}}$ is the membership matrix of $\mathbf{A}$ by solving Problem (5). Compared to Problem (1), Problem (5) is more amenable to analyze as it is the sum of separable equations. That is, for $i = 1, 2, \cdots, \frac{n(n-1)}{2}$, we can minimize each $\frac{1}{n}\|\mathbf{B}_{i\cdot} - \mathbf{Y}_{i\cdot}\|_2^2 + \alpha\|\mathbf{Y}_{i\cdot}\|_2$ individually with the additional constraint $\mathbf{\Omega}\mathbf{Y} = 0$. Following standard convex analysis (Page 303 of [2]), $\hat{\mathbf{Y}}$ and $\hat{\mathbf{\Lambda}}$ are an optimal primal and dual solution pair of Problem (5) if and only if

$$\hat{\mathbf{Y}}_{\cdot j} \in \mathbb{V}, (\hat{\mathbf{\Lambda}}_{\cdot j})^T \in \mathbb{V}^\perp, \ j = 1, 2, \cdots, p, \tag{6}$$

and

$$\hat{\mathbf{Y}}_{i\cdot} \in \arg\min_{\mathbf{y} \in \mathbb{R}^p}(\frac{1}{n}\|\mathbf{B}_{i\cdot} - \mathbf{y}\|_2^2 + \alpha\|\mathbf{y}\|_2 - \mathbf{y}\hat{\mathbf{\Lambda}}_{i\cdot}^T), \ i = 1, 2, \cdots, \binom{n}{2}. \tag{7}$$

**Step 2:** In this step, we construct $\hat{\mathbf{\Lambda}}$. Since $\mathbf{A}$ is constructed by concatenating matrices $\mathbf{A}^1$ and $\mathbf{A}^2$ vertically, we also expect $\hat{\mathbf{X}}$ to be concatenated by two matrices vertically. Due to the fact that $\hat{\mathbf{Y}} = \mathfrak{D}(\hat{\mathbf{X}})$, for $1 \le l \le p$, we write $\hat{\mathbf{Y}}$ and $\hat{\mathbf{\Lambda}}$ as the following

$$\hat{\mathbf{\Lambda}}_{\cdot l} = \begin{pmatrix} \hat{\mathbf{\Lambda}}_{\cdot l}^1 \\ \hat{\mathbf{\Lambda}}_{\cdot l}^2 \\ \hat{\mathbf{\Lambda}}_{\cdot l}^{(1,2)} \end{pmatrix} \text{ and } \hat{\mathbf{Y}}_{\cdot l} = \begin{pmatrix} \hat{\mathbf{Y}}_{\cdot l}^1 \\ \hat{\mathbf{Y}}_{\cdot l}^2 \\ \hat{\mathbf{Y}}_{\cdot l}^{(1,2)} \end{pmatrix}$$

where $\hat{\mathbf{\Lambda}}_{\cdot l}^i$, $\hat{\mathbf{Y}}_{\cdot l}^i \in \mathbb{R}^{\binom{n_i}{2}}$ for $i = 1, 2$ and $\hat{\mathbf{\Lambda}}_{\cdot l}^{(1,2)}$, $\hat{\mathbf{Y}}_{\cdot l}^{(1,2)} \in \mathbb{R}^{n_1 n_2}$, which are determined below.

By the structure of $\mathbf{\Omega}$, after some algebraic operations, it can be shown that $(\hat{\mathbf{\Lambda}}_{\cdot l})^T \in \mathbb{V}^\perp$ is equivalent to the following equalities that hold,

$$\mathbf{R}_{n_1}^T \hat{\mathbf{\Lambda}}_{\cdot l}^1 = -\mathbf{S}_{n_1 n_2 \times n_1}^T \hat{\mathbf{\Lambda}}_{\cdot l}^{(1,2)}, \quad \mathbf{R}_{n_2}^T \hat{\mathbf{\Lambda}}_{\cdot l}^2 = -\mathbf{W}_{n_1 n_2 \times n_2}^T \hat{\mathbf{\Lambda}}_{\cdot l}^{(1,2)}. \tag{8}$$

We now construct $\hat{\mathbf{\Lambda}}^{(1,2)}$. Set

$$\hat{\mathbf{\Lambda}}_{m\cdot}^{(1,2)} = \frac{2}{n} \left( \mathfrak{M}\left(\mathbf{B}^{(1,2)}\right) - \mathbf{B}_{m\cdot}^{(1,2)} \right), \ 1 \le m \le n_1 n_2. \tag{9}$$

Since $\hat{\mathbf{\Lambda}}^{(1,2)}$ is now fixed, we can bound the right hand sides of the two equalities in (8). In order to bound the entries of $\hat{\mathbf{\Lambda}}_{\cdot l}^1$ and $\hat{\mathbf{\Lambda}}_{\cdot l}^2$, we need the following lemma.

**Lemma 3.** *Given* $\mathbf{c}_n \in \mathbb{R}^n$*, i.e.,* $\mathbf{c}_n = (c_1, c_2, \cdots, c_n)^T$*, such that* $\sum_{i=1}^{n} c_i = 0$ *and* $\exists b \in \mathbb{R}, |c_i| \le b$*, then* $\exists \mathbf{x} \in \mathbb{R}^{\frac{n(n-1)}{2}}$*, such that* $\|\mathbf{x}\|_\infty \le \frac{2}{n}b$ *and* $\mathbf{R}_n^T \mathbf{x} = \mathbf{c}_n$*.*

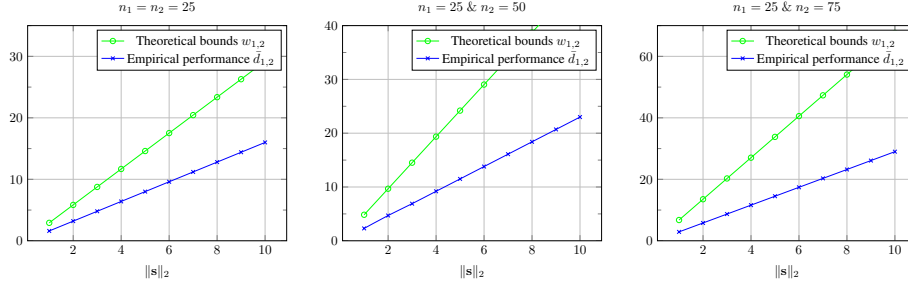

Figure 1: Theoretical bounds and empirical performance. This figure illustrates the case in which $n_1$, $n_2$ are constants and $\|\mathbf{s}\|_2$ is increasing.

Then, because we can bound the right hand sides of the two equalities of (8), by using Lemma 3, we can show that there exist $\hat{\mathbf{\Lambda}}^1_{\cdot l}, \hat{\mathbf{\Lambda}}^2_{\cdot l}$ satisfying (8) such that the following holds

$$\|\hat{\mathbf{\Lambda}}^1_{\cdot l}\|_\infty \leq \frac{2}{n}(n_2)\frac{(n_1-1)}{n_1^2}(4\sigma_{1l}) \text{ and } \|\hat{\mathbf{\Lambda}}^2_{\cdot l}\|_\infty \leq \frac{2}{n}(n_1)\frac{(n_2-1)}{n_2^2}(4\sigma_{2l}). \tag{10}$$

To summarize this step, we have constructed $\hat{\mathbf{\Lambda}}$ of dimension $\binom{n}{2} \times p$ such that

$$\begin{cases} \hat{\mathbf{\Lambda}}^1_{\cdot l}, \hat{\mathbf{\Lambda}}^2_{\cdot l} \text{ satisfies (10)}, \ 1 \leq l \leq p, \\ \hat{\mathbf{\Lambda}}^{(1,2)}_{m\cdot} = \frac{2}{n}\left(\mathfrak{M}\left(\mathbf{B}^{(1,2)}\right) - \mathbf{B}^{(1,2)}_{m\cdot}\right), \ 1 \leq m \leq n_1 n_2. \end{cases}$$

**Step 3:** Finally, we construct $\hat{\mathbf{Y}}$. Set

$$\begin{cases} \hat{\mathbf{Y}}^1_{\cdot l} = \hat{\mathbf{Y}}^2_{\cdot l} = 0, \ 1 \leq l \leq p, \\ \hat{\mathbf{Y}}^{(1,2)}_{m\cdot} = \left(1 - \dfrac{n\alpha}{2\|\mathfrak{M}\left(\mathbf{B}^{(1,2)}\right)\|_2}\right)\left(\mathfrak{M}\left(\mathbf{B}^{(1,2)}\right)\right), \ 1 \leq m \leq n_1 n_2. \end{cases}$$

Choosing $w_{1,2} < \frac{n}{2}\alpha < d_{1,2}$, according to $\hat{\mathbf{\Lambda}}$ and $\hat{\mathbf{Y}}$ constructed, it is easy to checked that conditions (6) and (7) are satisfied. So $\hat{\mathbf{\Lambda}}$ and $\hat{\mathbf{Y}}$ are an optimal primal and dual solution pair of Problem (5).

## 5  Experiments

We now report some numerical experimental results. The empirical performance of SON has been reported in numerous works [8, 10, 11]. It has been shown that SON outperforms traditional clustering methods like K-means in many situations. As such, we do not reproduce such results. Instead, we conduct experiments to validate our theoretic results.

Recall that Theorem 1 states that when samples are drawn from two cubes, SON guarantees to successfully recover the cluster membership if the distance between cubes is larger than a threshold which is linear to the cube size $\|\mathbf{s}_i\|$ and the ratio between $n_1$ and $n_2$. To validate this, we randomly draw a data matrix $\mathbf{A}$ where each row belongs to one of the two cubes, and find numerically the *largest* distance $\bar{d}_{1,2}$ between the cubes where the cluster membership is *not* correctly recovered. Clearly, $\bar{d}_{1,2}$ provides an empirical estimator of the minimal distance needed to successfully recover the cluster membership. We compare the theoretic bound $w_{1,2}$ with the empirical performance $\bar{d}_{1,2}$ to validate our theorem. The specific procedures of the experiments are as follows.

1. Choose two cubes $\mathbb{C}^1$ and $\mathbb{C}^2$ from space $\mathbb{R}^p$ with size $\mathbf{s}_1 = 2(\sigma_{11}, \sigma_{12}, \cdots, \sigma_{1p})$ and $\mathbf{s}_2 = 2(\sigma_{21}, \sigma_{22}, \cdots, \sigma_{2p})$, and the distance between $\mathbb{C}^1$ and $\mathbb{C}^2$ is $d$.

2. Choose arbitrarily $n_1$ points from $\mathbb{C}^1$ and $n_2$ points from $\mathbb{C}^2$ and form the data matrix $\mathbf{A}^d$ of dimension $n \times p$. Repeat and sample $m$ data matrices $\{\mathbf{A}^d_1, \mathbf{A}^d_2, \cdots, \mathbf{A}^d_m\}$.

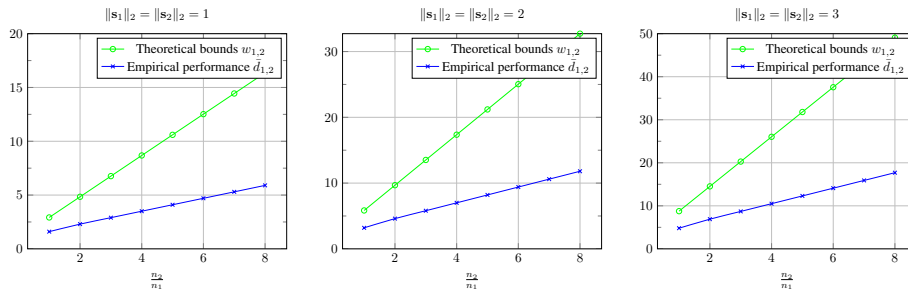

Figure 2: Theoretical bounds and empirical performance. This figure illustrates the case in which $\|\mathbf{s}_1\|_2, \|\mathbf{s}_2\|_2$ are constants and the ratio $\frac{n_2}{n_1}$ is increasing.

3. Repeat for different $d$. Set

$$\bar{d}_{1,2} = \max\{d | \exists 1 \le j \le m \text{ s.t. SON fails to determine the cluster membership of } \mathbf{A}_j^d\}.$$

4. Repeat for different cube sizes $\|\mathbf{s}_1\|_2$ and $\|\mathbf{s}_2\|_2$.
5. Repeat for different sample numbers $n_1$ and $n_2$.

In the experiments, we focus on the samples chosen from $\mathbb{R}^2$, i.e., $p = 2$, and use synthetic data to obtain the empirical performance. The results are shown in Figure 1 and 2. Figure 1 presents the situation where $n_1$ and $n_2$ are fixed and the cube sizes are increasing. In particular, the two cubes are both of size $l \times l$, i.e., both with edge length $(l, l)$. Thus we have $\|\mathbf{s}\|_2 = \sqrt{2}l$. Clearly, we can see that the empirical performance and the theoretical bounds are both linearly increasing with respect to $\|\mathbf{s}\|_2$, which implies that our theoretical results correctly predict how the performance of SON depends on $\|\mathbf{s}\|_2$. Figure 2 presents the situation in which $\|\mathbf{s}\|_1$ and $\|\mathbf{s}\|_2$ are fixed, while the ratio $\frac{n_2}{n_1}$ is changing. Again, we observe that both the empirical performance and the theoretical bounds are linearly increasing with respect to $\frac{n_2}{n_1}$, which implies that our theoretical bounds $w_{1,2}$ predict the correct relation between the performance of SON and $\frac{n_2}{n_1}$.

## 6 Conclusion

In this paper, we provided theoretical analysis for the recently presented convex optimization procedure for clustering, which we term as SON. We showed that if all samples are drawn from two clusters, each being a cube, then SON is guaranteed to successfully recover the cluster membership provided that the distance between the two cubes is greater than the "weighted size" – a term that linearly depends on the cube size and the ratio between the numbers of the samples in each cluster. Such linear dependence is also observed in our numerical experiment, which demonstrates (at least qualitatively) the validity of our results.

The main thrust of this paper is to explore using techniques from high-dimensional statistics, in particular regularization methods that extract low-dimensional structures such as sparsity or low-rankness, to tackle clustering problems. These techniques have recently been successfully applied to graph clustering and subspace clustering [4, 7, 12, 5, 9], but not so much to distance-based clustering tasks with the only exception of SON, to the best of our knowledge. This paper is the first attempt to provide a rigorous analysis to derive sufficient conditions when SON succeeds. We believe this not only helps to understand why SON works in practice as shown in previous works [8, 10, 11], but also provides important insights to develop novel algorithms based on high-dimensional statistics tools for clustering tasks.

**Acknowledgments**

The work of H. Xu was partially supported by the Ministry of Education of Singapore through AcRF Tier Two grant R-265-000-443-112. This work is also partially supported by the grant from Microsoft Research Asia with grant number R-263-000-B13-597.

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
