[Supplementary Material]

# Convex Optimization Procedure for Clustering: Theoretical Revisit Supplementary Material

**Changbo Zhu**
Department of Electrical and Computer Engineering
Department of Mathematics
National University of Singapore
elezhuc@nus.edu.sg

**Huan Xu**
Department of Mechanical Engineering
National University of Singapore
mpexuh@nus.edu.sg

**Chenlei Leng**
Department of Statistics
University of Warwick
c.leng@warwick.ac.uk

**Shuicheng Yan**
Department of Electrical and Computer Engineering
National University of Singapore
eleyans@nus.edu.sg

## 1 SON

Recall that we are analysing the following convex optimization problem, which we term as SON

$$\hat{\mathbf{X}} = \arg \min_{\mathbf{X} \in \mathbb{R}^{n \times p}} \|\mathbf{A} - \mathbf{X}\|_F^2 + \alpha \sum_{i<j} \|\mathbf{X}_{i\cdot} - \mathbf{X}_{j\cdot}\|_2, \tag{1}$$

where $\mathbf{A}$ is a given data matrix of dimension $n \times p$ such that each row is a data point, $\alpha$ is a tunable parameter, $\| \cdot \|_F$ denotes the Frobenius norm and $\mathbf{X}_{i\cdot}$ denotes the $i$th row of $\mathbf{X}$.

## 2 Proof of Lemma 1

**Lemma 1.** *If the data matrix $\mathbf{A}$ is column centered, then the optimal solution $\hat{\mathbf{X}}$ of problem (1) is also column centered. Further more, set $\mathbf{B} = \mathfrak{D}(\mathbf{A})$ and $\hat{\mathbf{Y}} = \mathfrak{D}(\hat{\mathbf{X}})$, we have*

$$\|\mathbf{A} - \hat{\mathbf{X}}\|_F^2 = \sum_{i=1}^{\frac{n(n-1)}{2}} \frac{1}{n} \|\mathbf{B}_{i\cdot} - \hat{\mathbf{Y}}_{i\cdot}\|_2^2.$$

*Proof.* For any $\mathbf{X} \in \mathbb{R}^{n \times p}$, set $\mathbf{Y} = \mathfrak{D}(\mathbf{X})$, we have

$$\sum_{k=1}^{p} (\mathbf{X}_{1,k} + \cdots + \mathbf{X}_{n,k})(\mathbf{X}_{1,k} + \cdots + \mathbf{X}_{n,k}) = \sum_{k=1}^{p} \left( \sum_{i=1}^{n} \mathbf{X}_{i,k}^2 + 2 \sum_{1 \le i < j \le n} \mathbf{X}_{i,k}\mathbf{X}_{j,k} \right),$$

which implies that

$$2 \sum_{k=1}^{p} \sum_{1 \le i < j \le n} \mathbf{X}_{i,k}\mathbf{X}_{j,k} = \sum_{k=1}^{p} \left( (\mathbf{X}_{1,k} + \cdots + \mathbf{X}_{n,k})^2 - \sum_{i=1}^{n} \mathbf{X}_{i,k}^2 \right).$$

Similarly, we have

$$2 \sum_{k=1}^{p} \sum_{1 \le i < j \le n} \mathbf{A}_{i,k}\mathbf{A}_{j,k} = \sum_{k=1}^{p} \left( (\mathbf{A}_{1,k} + \cdots + \mathbf{A}_{n,k})^2 - \sum_{i=1}^{n} \mathbf{A}_{i,k}^2 \right).$$

Then, because $\mathbf{A}$ is column centered, we get

$$0 = \sum_{k=1}^{p} \left(\mathbf{A}_{1,k} + \cdots + \mathbf{A}_{n,k}\right) \left(\mathbf{X}_{1,k} + \cdots + \mathbf{X}_{n,k}\right)$$

$$= \sum_{k=1}^{p} \left( \sum_{1 \le i < j \le n} \left(\mathbf{X}_{i,k}\mathbf{A}_{j,k} + \mathbf{A}_{i,k}\mathbf{X}_{j,k}\right) + \sum_{i=1}^{n} \mathbf{A}_{i,k}\mathbf{X}_{i,k} \right),$$

which implies directly that

$$2\sum_{k=1}^{p} \sum_{1 \le i < j \le n} \left(-\mathbf{X}_{i,k}\mathbf{A}_{j,k} - \mathbf{A}_{i,k}\mathbf{X}_{j,k}\right)$$

$$= \sum_{k=1}^{p} \left(-2(\mathbf{X}_{1,k} + \cdots + \mathbf{X}_{n,k})(\mathbf{A}_{1,k} + \cdots + \mathbf{A}_{n,k}) + 2\sum_{i=1}^{n}\mathbf{A}_{i,k}\mathbf{X}_{i,k}\right).$$

Next, we can see that

$$2\sum_{k=1}^{p} \sum_{1 \le i < j \le n} \left(\mathbf{X}_{i,k}\mathbf{X}_{j,k} + \mathbf{A}_{i,k}\mathbf{A}_{j,k} - \mathbf{X}_{i,k}\mathbf{A}_{j,k} - \mathbf{A}_{i,k}\mathbf{X}_{j,k}\right)$$

$$= \sum_{k=1}^{p} \left((\mathbf{X}_{1,k} + \cdots + \mathbf{X}_{n,k})^2 - \sum_{i=1}^{n}\mathbf{X}_{i,k}^2\right)$$

$$+ \sum_{k=1}^{p} \left((\mathbf{A}_{1,k} + \cdots + \mathbf{A}_{n,k})^2 - \sum_{i=1}^{n}\mathbf{A}_{i,k}^2\right)$$

$$+ \sum_{k=1}^{p} \left(-2(\mathbf{X}_{1,k} + \cdots + \mathbf{X}_{n,k})(\mathbf{A}_{1,k} + \cdots + \mathbf{A}_{n,k}) + 2\sum_{i=1}^{n}\mathbf{A}_{i,k}\mathbf{X}_{i,k}\right)$$

$$= \sum_{k=1}^{p} \left(-\sum_{i=1}^{n}\mathbf{X}_{i,k}^2 - \sum_{i=1}^{n}\mathbf{A}_{i,k}^2 + 2\sum_{i=1}^{n}\mathbf{A}_{i,k}\mathbf{X}_{i,k}\right) + \sum_{k=1}^{p}(\mathbf{X}_{1,k} + \cdots + \mathbf{X}_{n,k})^2$$

$$= -\|\mathbf{A} - \mathbf{X}\|_F^2 + \sum_{k=1}^{p}(\mathbf{X}_{1,k} + \cdots + \mathbf{X}_{n,k})^2.$$

So, we get the following identity

$$\sum_{i=1}^{\frac{n(n-1)}{2}} \|\mathbf{B}_{i\cdot} - \mathbf{Y}_{i\cdot}\|_2^2$$

$$= \sum_{k=1}^{p} \sum_{1 \le i < j \le n} \left((\mathbf{X}_{i,k} - \mathbf{X}_{j,k}) - (\mathbf{A}_{i,k} - \mathbf{A}_{j,k})\right)^2$$

$$= \sum_{k=1}^{p} \sum_{1 \le i < j \le n} \left((\mathbf{X}_{i,k} - \mathbf{A}_{i,k})^2 + (\mathbf{X}_{j,k} - \mathbf{A}_{j,k})^2 - 2(\mathbf{X}_{i,k} - \mathbf{A}_{i,k})(\mathbf{X}_{j,k} - \mathbf{A}_{j,k})\right)$$

$$= \sum_{k=1}^{p}\sum_{i=1}^{n}(n-1)(\mathbf{X}_{i,k} - \mathbf{A}_{i,k})^2 - 2\sum_{k=1}^{p} \sum_{1 \le i < j \le n} \left(\mathbf{X}_{i,k}\mathbf{X}_{j,k} + \mathbf{A}_{i,k}\mathbf{A}_{j,k} - \mathbf{X}_{i,k}\mathbf{A}_{j,k} - \mathbf{A}_{i,k}\mathbf{X}_{j,k}\right)$$

$$= (n-1)\|\mathbf{A} - \mathbf{X}\|_F^2 - 2\sum_{k=1}^{p} \sum_{1 \le i < j \le n} \left(\mathbf{X}_{i,k}\mathbf{X}_{j,k} + \mathbf{A}_{i,k}\mathbf{A}_{j,k} - \mathbf{X}_{i,k}\mathbf{A}_{j,k} - \mathbf{A}_{i,k}\mathbf{X}_{j,k}\right)$$

$$= n\|\mathbf{A} - \mathbf{X}\|_F^2 - \sum_{k=1}^{p}(\mathbf{X}_{1,k} + \cdots + \mathbf{X}_{n,k})^2.$$

Then, we have

$$\|\mathbf{A} - \mathbf{X}\|_F^2 + \alpha \sum_{i<j} \| \mathbf{X}_{i\cdot} - \mathbf{X}_{j\cdot}\|_2$$

$$= \sum_{i=1}^{\frac{n(n-1)}{2}} \left( \frac{1}{n}\|\mathbf{B}_{i\cdot} - \mathbf{Y}_{i\cdot}\|_2^2 + \alpha\|\mathbf{Y}_{i\cdot}\|_2 \right) + \frac{1}{n}\sum_{j=1}^{p} \left( \sum_{i=1}^{n}\mathbf{X}_{i,j} \right)^2$$

$$\geq \sum_{i=1}^{\frac{n(n-1)}{2}} \left( \frac{1}{n}\|\mathbf{B}_{i\cdot} - \mathbf{Y}_{i\cdot}\|_2^2 + \alpha\|\mathbf{Y}_{i\cdot}\|_2 \right),$$

where the equality holds if and only if $\mathbf{X}$ is column centered.

Next, we prove that $\hat{\mathbf{X}}$ is column centered by contradiction. Suppose that $\hat{\mathbf{X}}$ is not column centered, then we can find a columned centered $\bar{\mathbf{X}} \in \mathbb{R}^{n \times p}$ s.t. $\hat{\mathbf{Y}} = \mathfrak{D}(\hat{\mathbf{X}}) = \mathfrak{D}(\bar{\mathbf{X}})$. Then, we have

$$\|\mathbf{A} - \hat{\mathbf{X}}\|_F^2 + \alpha \sum_{i<j} \|\hat{\mathbf{X}}_{i\cdot} - \hat{\mathbf{X}}_{j\cdot}\|_2$$

$$= \sum_{i=1}^{\frac{n(n-1)}{2}} \left( \frac{1}{n}\|\mathbf{B}_{i\cdot} - \hat{\mathbf{Y}}_{i\cdot}\|_2^2 + \alpha\|\hat{\mathbf{Y}}_{i\cdot}\|_2 \right) + \frac{1}{n}\sum_{j=1}^{p} \left( \sum_{i=1}^{n}\hat{\mathbf{X}}_{i,j} \right)^2$$

$$> \sum_{i=1}^{\frac{n(n-1)}{2}} \left( \frac{1}{n}\|\mathbf{B}_{i\cdot} - \hat{\mathbf{Y}}_{i\cdot}\|_2^2 + \alpha\|\hat{\mathbf{Y}}_{i\cdot}\|_2 \right)$$

$$= \|\mathbf{A} - \bar{\mathbf{X}}\|_F^2 + \alpha \sum_{i<j} \|\bar{\mathbf{X}}_{i\cdot} - \bar{\mathbf{X}}_{j\cdot}\|_2,$$

which contradicts the optimality of $\hat{\mathbf{X}}$. When $\hat{\mathbf{X}}$ is column centered, the following identity follows easily

$$n\|\mathbf{A} - \hat{\mathbf{X}}\|_F^2 = \sum_{i=1}^{\frac{n(n-1)}{2}} \|\mathbf{B}_{i\cdot} - \hat{\mathbf{Y}}_{i\cdot}\|_2^2 + \sum_{k=1}^{p}(\hat{\mathbf{X}}_{1,k} + \cdots + \hat{\mathbf{X}}_{n,k})^2$$

$$= \sum_{i=1}^{\frac{n(n-1)}{2}} \|\mathbf{B}_{i\cdot} - \hat{\mathbf{Y}}_{i\cdot}\|_2^2.$$

$\square$

**Remark:** following directly from the proof of Lemma 1, for any column centered matrices $\mathbf{G}$ and $\mathbf{H}$ in space $\mathbb{R}^{n \times p}$. Set $\tilde{\mathbf{G}} = \mathfrak{D}(\mathbf{G}), \tilde{\mathbf{H}} = \mathfrak{D}(\mathbf{H})$ we have

$$\|\mathbf{G} - \mathbf{H}\|_F^2 = \sum_{i=1}^{\frac{n(n-1)}{2}} \frac{1}{n}\|\tilde{\mathbf{G}}_{i\cdot} - \tilde{\mathbf{H}}_{i\cdot}\|_2^2.$$

# 3 Proof of Lemma 2

**Lemma 2.** *Given a column centered data matrix* $\mathbf{A}$*, set* $\mathbf{B} = \mathfrak{D}(\mathbf{A})$ *and* $\mathbb{S} = \{\mathbf{Z} \in \mathbb{R}^{\binom{n}{2} \times p} \mid \mathbf{\Omega}\mathbf{Z}_{\cdot j} = 0,\ 1 \leq j \leq p\}$*. Then, we have*

$$\hat{\mathbf{X}} = \arg \min_{\mathbf{X} \in \mathbb{R}^{n \times p}} \|\mathbf{A} - \mathbf{X}\|_F^2 + \alpha \sum_{i<j} \|\mathbf{X}_{i\cdot} - \mathbf{X}_{j\cdot}\|_2 \qquad (2)$$

$$\Longleftrightarrow \mathfrak{D}(\hat{\mathbf{X}}) = \arg \min_{\mathbf{Y} \in \mathbb{S}} \sum_{i=1}^{\frac{n(n-1)}{2}} \left( \frac{1}{n}\|\mathbf{B}_{i\cdot} - \mathbf{Y}_{i\cdot}\|_2^2 + \alpha\|\mathbf{Y}_{i\cdot}\|_2 \right). \qquad (3)$$

*Proof.* Set $\mathbb{T} = \{\mathbf{X} \in \mathbb{R}^{n \times p} | \mathbf{X} \text{ is column centered}\}$. Because $\mathbf{A}$ is column centered, we have that the optimal solution $\hat{\mathbf{X}}$ is also column centered by Lemma 1. So, we get

$$\hat{\mathbf{X}} = \arg \min_{\mathbf{X} \in \mathbb{R}^{n \times p}} \|\mathbf{A} - \mathbf{X}\|_F^2 + \alpha \sum_{i<j} \|\mathbf{X}_{i.} - \mathbf{X}_{j.}\|_2$$

$$\Longleftrightarrow \hat{\mathbf{X}} = \arg \min_{\mathbf{X} \in \mathbb{T}} \|\mathbf{A} - \mathbf{X}\|_F^2 + \alpha \sum_{i<j} \|\mathbf{X}_{i.} - \mathbf{X}_{j.}\|_2.$$

Set $\hat{\mathbf{Y}} = \mathfrak{D}(\hat{\mathbf{X}})$, then again by Lemma 1, we have the following equality

$$\|\mathbf{A} - \hat{\mathbf{X}}\|_F^2 + \alpha \sum_{i<j} \|\hat{\mathbf{X}}_{i.} - \hat{\mathbf{X}}_{j.}\|_2 = \sum_{i=1}^{\frac{n(n-1)}{2}} \left( \frac{1}{n} \|\mathbf{B}_{i.} - \hat{\mathbf{Y}}_{i.}\|_2^2 + \alpha \|\hat{\mathbf{Y}}_{i.}\|_2 \right).$$

Indeed, it is this identity that gives us a hint to consider the following problem instead,

$$\hat{\mathbf{Y}} = \arg \min_{\mathbf{Y}} \sum_{i=1}^{\frac{n(n-1)}{2}} \left( \frac{1}{n} \|\mathbf{B}_{i.} - \mathbf{Y}_{i.}\|_2^2 + \alpha \|\mathbf{Y}_{i.}\|_2 \right),$$

where $\mathbf{Y}$ can not take values from the whole space $\mathbb{R}^{\binom{n}{2} \times p}$, because we need $\mathbf{Y} = \mathfrak{D}(\mathbf{X}^*)$ for some $\mathbf{X}^* \in \mathbb{T}$. So, we defined some special matrices in Section 4 to indicate that $\mathbf{Y} = \mathfrak{D}(\mathbf{X}^*)$.

By the definition of $\boldsymbol{\Omega}$ and direct checking, we know that

$$\mathbf{Y} = \mathfrak{D}(\mathbf{X}^*) \text{ for some } \mathbf{X}^* \in \mathbb{T} \Longleftrightarrow \mathbf{Y} \in \mathbb{S}.$$

Next, set $\hat{\mathbf{Y}} = \mathfrak{D}(\hat{\mathbf{X}})$, we show that $\hat{\mathbf{X}}$ is the optimal solution of problem (2) if and only if $\hat{\mathbf{Y}}$ is the optimal solution of problem (3). For all $\mathbf{Y}' \in \mathbb{S}$, $\exists \mathbf{X}' \in \mathbb{T}$ s.t. $\mathfrak{D}(\mathbf{X}') = \mathbf{Y}'$, so we have

$$\sum_{i=1}^{\frac{n(n-1)}{2}} \left( \frac{1}{n} \|\mathbf{B}_{i.} - \mathbf{Y}'_{i.}\|_2^2 + \alpha \|\mathbf{Y}'_{i.}\|_2 \right)$$

$$= \|\mathbf{A} - \mathbf{X}'\|_F^2 + \alpha \sum_{i<j} \|\mathbf{X}'_{i.} - \mathbf{X}'_{j.}\|_2$$

$$\geq \|\mathbf{A} - \hat{\mathbf{X}}\|_F^2 + \alpha \sum_{i<j} \|\hat{\mathbf{X}}_{i.} - \hat{\mathbf{X}}_{j.}\|_2$$

$$= \sum_{i=1}^{\frac{n(n-1)}{2}} \left( \frac{1}{n} \|\mathbf{B}_{i.} - \hat{\mathbf{Y}}_{i.}\|_2^2 + \alpha \|\hat{\mathbf{Y}}_{i.}\|_2 \right),$$

so, we have

$$\hat{\mathbf{Y}} = \arg \min_{\mathbf{Y} \in \mathbb{S}} \sum_{i=1}^{\frac{n(n-1)}{2}} \left( \frac{1}{n} \|\mathbf{B}_{i.} - \mathbf{Y}_{i.}\|_2^2 + \alpha \|\mathbf{Y}_{i.}\|_2 \right).$$

On the contrary, suppose we are given that

$$\bar{\mathbf{Y}} = \arg \min_{\mathbf{Y} \in \mathbb{S}} \sum_{i=1}^{\frac{n(n-1)}{2}} \left( \frac{1}{n} \|\mathbf{B}_{i.} - \mathbf{Y}_{i.}\|_2^2 + \alpha \|\mathbf{Y}_{i.}\|_2 \right).$$

Then, $\exists \bar{\mathbf{X}} \in \mathbb{T}$ s.t. $\mathfrak{D}(\bar{\mathbf{X}}) = \bar{\mathbf{Y}}$. $\forall \mathbf{X}' \in \mathbb{T}$, denote $\mathbf{Y}' = \mathfrak{D}(\mathbf{X}')$, we have

$$\|\mathbf{A} - \mathbf{X}'\|_F^2 + \alpha \sum_{i<j} \|\mathbf{X}'_{i\cdot} - \mathbf{X}'_{j\cdot}\|_2$$

$$= \sum_{i=1}^{\frac{n(n-1)}{2}} \left( \frac{1}{n} \|\mathbf{B}_{i\cdot} - \mathbf{Y}'_{i\cdot}\|_2^2 + \alpha \|\mathbf{Y}'_{i\cdot}\|_2 \right)$$

$$\geq \sum_{i=1}^{\frac{n(n-1)}{2}} \left( \frac{1}{n} \|\mathbf{B}_{i\cdot} - \bar{\mathbf{Y}}_{i\cdot}\|_2^2 + \alpha \|\bar{\mathbf{Y}}_{i\cdot}\|_2 \right)$$

$$= \|\mathbf{A} - \bar{\mathbf{X}}\|_F^2 + \alpha \sum_{i<j} \|\bar{\mathbf{X}}_{i\cdot} - \bar{\mathbf{X}}_{j\cdot}\|_2,$$

so, we get

$$\bar{\mathbf{X}} = \arg\min_{\mathbf{X} \in \mathbb{T}} \|\mathbf{A} - \mathbf{X}\|_F^2 + \alpha \sum_{i<j} \|\mathbf{X}_{i\cdot} - \mathbf{X}_{j\cdot}\|_2.$$

In conclusion, we have showed the following result

$$\hat{\mathbf{X}} = \arg\min_{\mathbf{X} \in \mathbb{T}} \|\mathbf{A} - \mathbf{X}\|_F^2 + \alpha \sum_{i<j} \|\mathbf{X}_{i\cdot} - \mathbf{X}_{j\cdot}\|_2$$

$$\Longleftrightarrow \mathfrak{D}(\hat{\mathbf{X}}) = \arg\min_{\mathbf{Y} \in \mathbb{S}} \sum_{i=1}^{\frac{n(n-1)}{2}} \left( \frac{1}{n} \|\mathbf{B}_{i\cdot} - \mathbf{Y}_{i\cdot}\|_2^2 + \alpha \|\mathbf{Y}_{i\cdot}\|_2 \right).$$

$\square$

## 4 Proof of Lemma 3

**Lemma 3.** *Given* $\mathbf{c}_n \in \mathbb{R}^n$, *i.e.* $\mathbf{c}_n = (c_1, c_2, \cdots, c_n)^T$, *s.t.* $\sum_{i=1}^{n} c_i = 0$ *and* $\exists b \in \mathbb{R}, |c_i| \leq b$. *Then* $\exists \mathbf{x} \in \mathbb{R}^{\frac{n(n-1)}{2}}$, *s.t.* $\|\mathbf{x}\|_\infty \leq \frac{2}{n} b$ *and* $\mathbf{R}_n^T \mathbf{x} = \mathbf{c}_n$.

*Proof.* Set

$$\mathbb{F} = \left\{ (x_1, x_2, \cdots, x_n)^T \in \mathbb{R}^n | \sum_{i=1}^{n} x_i = 0, 1 \leq j \leq n, |x_j| \leq b \right\},$$

and

$$\mathbb{G} = \left\{ \left( x_1, x_2, \cdots, x_{\frac{n(n-1)}{2}} \right)^T \in \mathbb{R}^{\frac{n(n-1)}{2}} | 1 \leq i \leq \frac{n(n-1)}{2}, |x_i| \leq \frac{2}{n} b \right\}.$$

Notice that $\mathbb{F}$ is convex and define $f : \mathbb{R}^{\frac{n(n-1)}{2}} \mapsto \mathbb{R}^n$ as $f(\mathbf{x}) = \mathbf{R}_n^T \mathbf{x}$ for any $\mathbf{x} \in \mathbb{R}^{\frac{n(n-1)}{2}}$. Then, we want to show that for all $\mathbf{c}_n \in \mathbb{F}$, exists $\mathbf{x} \in \mathbb{G}$ such that $f(\mathbf{x}) = \mathbf{c}_n$. Equivalently, we want to show $f(\mathbb{G}) \supseteq \mathbb{F}$.

Let $\mathbf{y} = (y_1, y_2, \cdots, y_n)^T \in \mathbb{R}^n$. If $n$ is even, set $y_1 = y_2 = \cdots = y_{\frac{n}{2}} = b$ and $y_{\frac{n}{2}+1} = y_{\frac{n}{2}+2} = \cdots = y_n = -b$. If $n$ is odd, set $y_1 = y_2 = \cdots = y_{\frac{n-1}{2}} = b$, $y_{\frac{n-1}{2}+1} = y_{\frac{n-1}{2}+2} = \cdots = y_{n-1} = -b$ and $y_n = 0$. Then, let $\mathbb{P}_n$ denote the set of all permutations $\mathbf{p}$ of the sequence of integers $\{1, 2, \cdots, n\}$. After that, let $\mathbb{E}$ denote the set of all extreme points of the convex set $\mathbb{F}$, then it is easy to see that

$$\mathbb{E} = \left\{ (z_1, z_2, \cdots, z_n)^T \in \mathbb{R}^n | \exists \mathbf{p} \in \mathbb{P}_n \text{ s.t. } \left( z_{\mathbf{p}(1)}, z_{\mathbf{p}(2)}, \cdots, z_{\mathbf{p}(n)} \right)^T = \mathbf{y} \right\}.$$

In the following, we show that $\mathbb{E} \subseteq f(\mathbb{G})$. Given any $\mathbf{z}_n = (z_1, z_2, \cdots, z_n)^T \in \mathbb{E}$ we construct a $\mathbf{u} \in \mathbb{G}$ s.t. $\mathbf{R}_n^T \mathbf{u} = \mathbf{z}_n$.

Denote
$$\mathbf{u} = \left(u_1^{n-1}, u_2^{n-1}, \cdots, u_{n-1}^{n-1}, u_1^{n-2}, \cdots, u_{n-2}^{n-2}, \cdots, u_1^1\right)^T.$$

For $1 \le i < j \le n$, when $n$ is even set
$$u_j^i = \begin{cases} \frac{2}{n}b & : z_{n-i} > z_{n-i+j} \\ -\frac{2}{n}b & : z_{n-i} < z_{n-i+j} \\ 0 & : z_{n-i} = z_{n-i+j} \end{cases}$$

and when $n$ is odd set
$$u_j^i = \begin{cases} \frac{2}{n+1}b & : z_{n-i} > z_{n-i+j} \\ -\frac{2}{n+1}b & : z_{n-i} < z_{n-i+j} \\ 0 & : z_{n-i} = z_{n-i+j} \end{cases}$$

By this construction, checking directly that $\mathbf{u} \in \mathbb{G}$ and $\mathbf{R}_n^T \mathbf{u} = \mathbf{z}_n$. So, we have $\mathbb{E} \subseteq f(\mathbb{G})$. Next, since $f$ is an affine function and the image of a convex set under an affine function is convex, we have $f(\mathbb{G})$ is convex. So, we have $\mathbb{F} = \{\text{convex hull of } \mathbb{E}\} \subseteq f(\mathbb{G})$. $\qquad\square$

## 5 Proof of Theorem 1

**Theorem 1.** *Given a column centered data matrix $\mathbf{A}$ of dimension $n \times p$, where each row is arbitrarily picked from either cube $\mathbb{C}^1$ or cube $\mathbb{C}^2$ and there are totally $n_i$ rows chosen from $\mathbb{C}^i$ for $i = 1, 2$, if $w_{1,2} < d_{1,2}$, then by choosing the parameter $\alpha \in \mathbb{R}$ such that $w_{1,2} < \frac{n}{2}\alpha < d_{1,2}$, we have the following:*

1. *SON can correctly determine the cluster membership of $\mathbf{A}$;*

2. *Rearrange the rows of $\mathbf{A}$ such that*

$$\mathbf{A} = \begin{pmatrix} \mathbf{A}^1 \\ \mathbf{A}^2 \end{pmatrix} \text{ and } \mathbf{A}^i = \begin{pmatrix} \mathbf{A}_{1.}^i \\ \mathbf{A}_{2.}^i \\ \vdots \\ \mathbf{A}_{n_i.}^i \end{pmatrix}, \qquad (4)$$

*where for $i = 1, 2$ and $j = 1, 2, \cdots, n_i$, $\mathbf{A}_{j.}^i = (\mathbf{A}_{j,1}^i, \mathbf{A}_{j,2}^i, \cdots, \mathbf{A}_{j,p}^i) \in \mathbb{C}^i$. Then, the optimal solution $\hat{\mathbf{X}}$ of problem (1) is given by*

$$\hat{\mathbf{X}}_{i.} = \begin{cases} \frac{n_2}{n_1+n_2}\left(1 - \frac{n\alpha}{2\|\mathfrak{M}(\mathfrak{D}_2(\mathbf{A}^1,\mathbf{A}^2))\|_2}\right)\mathfrak{M}\left(\mathfrak{D}_2(\mathbf{A}^1,\mathbf{A}^2)\right), & \text{if } \mathbf{A}_{i.} \in \mathbb{C}^1; \\ -\frac{n_1}{n_1+n_2}\left(1 - \frac{n\alpha}{2\|\mathfrak{M}(\mathfrak{D}_2(\mathbf{A}^1,\mathbf{A}^2))\|_2}\right)\mathfrak{M}\left(\mathfrak{D}_2(\mathbf{A}^1,\mathbf{A}^2)\right), & \text{if } \mathbf{A}_{i.} \in \mathbb{C}^2. \end{cases}$$

*Proof.* WLOG, we let
$$\mathbf{A} = \begin{pmatrix} \mathbf{A}^1 \\ \mathbf{A}^2 \end{pmatrix} \text{ and } \mathbf{A}^i = \begin{pmatrix} \mathbf{A}_{1.}^i \\ \mathbf{A}_{2.}^i \\ \vdots \\ \mathbf{A}_{n_i.}^i \end{pmatrix},$$

where for $i = 1, 2$ and $j = 1, 2, \cdots, n_i$, $\mathbf{A}_{j.}^i = (\mathbf{A}_{j,1}^i, \mathbf{A}_{j,2}^i, \cdots, \mathbf{A}_{j,p}^i) \in \mathbb{C}^i$.

**Step 1:** In this step, we derive an equivalent form of problem (1) and give optimality conditions. For convenience, set $\mathbf{B}^{(1,2)} = \mathfrak{D}_2(\mathbf{A}^1, \mathbf{A}^2)$, $\mathbf{B}^1 = \mathfrak{D}_1(\mathbf{A}^1)$, $\mathbf{B}^2 = \mathfrak{D}_1(\mathbf{A}^2)$, $\mathbb{V} = \{\mathbf{y} \in \mathbb{R}^{\binom{n}{2}} \mid \mathbf{\Omega}\mathbf{y} = 0\}$ and $\mathbb{S} = \{\mathbf{Z} \in \mathbb{R}^{\binom{n}{2} \times p} \mid \mathbf{\Omega}\mathbf{Z}_{.j} = 0, \ 1 \le j \le p\}$. Due to lemma (2), we can focus on the following problem

$$\hat{\mathbf{Y}} = \arg\min_{\mathbf{Y} \in \mathbb{S}} \sum_{i=1}^{\frac{n(n-1)}{2}} \left(\frac{1}{n}\|\mathbf{B}_{i.} - \mathbf{Y}_{i.}\|_2^2 + \alpha\|\mathbf{Y}_{i.}\|_2\right). \qquad (5)$$

We use $\hat{\boldsymbol{\Lambda}}$ to denote the optimal dual solution of problem (5) which has the same dimension as $\hat{\mathbf{Y}}$. Then, by Proposition 6.4.3 in [1] Page 303, we have the following result, $\hat{\mathbf{Y}}$ and $\hat{\boldsymbol{\Lambda}}$ are an optimal primal and dual solution pair of (5) if and only if

$$\hat{\mathbf{Y}}_{\cdot j} \in \mathbb{V}, (\hat{\boldsymbol{\Lambda}}_{\cdot j})^T \in \mathbb{V}^\perp, \ j = 1, 2, \cdots, p, \tag{6}$$

and

$$\hat{\mathbf{Y}}_{i\cdot} \in \arg\min_{\mathbf{y} \in \mathbb{R}^p} \left( \frac{1}{n} \|\mathbf{B}_{i\cdot} - \mathbf{y}\|_2^2 + \alpha \|\mathbf{y}\|_2 - \mathbf{y}\hat{\boldsymbol{\Lambda}}_{i\cdot}^T \right), \ i = 1, 2, \cdots, \binom{n}{2}. \tag{7}$$

**Step 2:** In this step, we construct $\hat{\boldsymbol{\Lambda}}$. Since $\mathbf{A}$ is constructed by concatenating matrices $\mathbf{A}^1$ and $\mathbf{A}^2$ vertically, we also expect $\hat{\mathbf{X}}$ to be concatenated by two matrices vertically. Due to the fact that $\hat{\mathbf{Y}} = \mathfrak{D}(\hat{\mathbf{X}})$, for $1 \leq l \leq p$, we write $\hat{\mathbf{Y}}$ and $\hat{\boldsymbol{\Lambda}}$ as the following

$$\hat{\boldsymbol{\Lambda}}_{\cdot l} = \begin{pmatrix} \hat{\boldsymbol{\Lambda}}_{\cdot l}^1 \\ \hat{\boldsymbol{\Lambda}}_{\cdot l}^2 \\ \hat{\boldsymbol{\Lambda}}_{\cdot l}^{(1,2)} \end{pmatrix} \text{ and } \hat{\mathbf{Y}}_{\cdot l} = \begin{pmatrix} \hat{\mathbf{Y}}_{\cdot l}^1 \\ \hat{\mathbf{Y}}_{\cdot l}^2 \\ \hat{\mathbf{Y}}_{\cdot l}^{(1,2)} \end{pmatrix}$$

where $\hat{\boldsymbol{\Lambda}}_{\cdot l}^i$, $\hat{\mathbf{Y}}_{\cdot l}^i \in \mathbb{R}^{\binom{n_i}{2}}$ for $i = 1, 2$ and $\hat{\boldsymbol{\Lambda}}_{\cdot l}^{(1,2)}$, $\hat{\mathbf{Y}}_{\cdot l}^{(1,2)} \in \mathbb{R}^{n_1 n_2}$. Next, we use $\text{Row}(\boldsymbol{\Omega})$ to denote the row space of $\boldsymbol{\Omega}$, then $\mathbb{V}^T$ is the same as $\text{Row}(\boldsymbol{\Omega})$.

For notational convenience, given any vector $\mathbf{v}$, we use $\mathbf{v}[i, j], i < j$ to denote a new vector composed of the $i$th through $j$th element of $\mathbf{v}$. By the structure of $\boldsymbol{\Omega}$, i.e. there exists a identity submatrix $\mathbf{I}$ of $\boldsymbol{\Omega}$ s.t. $\mathbf{I}$ and $\boldsymbol{\Omega}$ have the same number of rows, we have $(\hat{\boldsymbol{\Lambda}}_{\cdot l})^T \in \text{Row}(\boldsymbol{\Omega})$ is equivalent to the following equalities ,i.e. equalities (8) and (9),

$$\mathbf{R}_{n_1-1}^T \left( \hat{\boldsymbol{\Lambda}}_{\cdot l}^1 \left[ n_1 : \binom{n_1}{2} \right] \right) + \mathbf{S}_{n_2(n_1-1)\times(n_1-1)}^T \left( \hat{\boldsymbol{\Lambda}}_{\cdot l}^{(1,2)}[n_2 + 1 : n_1 n_2] \right) = \hat{\boldsymbol{\Lambda}}_{\cdot l}^1 [1 : n_1 - 1], \tag{8}$$

$$\mathbf{R}_{n_2}^T \hat{\boldsymbol{\Lambda}}_{\cdot l}^2 + \mathbf{W}_{(n_1-1)n_2 \times n_2}^T \hat{\boldsymbol{\Lambda}}_{\cdot l}^{(1,2)}[n_2 + 1 : n_1 n_2] = \hat{\boldsymbol{\Lambda}}_{\cdot l}^{(1,2)} [1 : n_2]. \tag{9}$$

Then, we set

$$\hat{\boldsymbol{\Lambda}}_{m\cdot}^{(1,2)} = \frac{2}{n} \left( \frac{1}{n_1 n_2} \left( \sum_{k=1}^{n_1 n_2} \mathbf{B}_{k\cdot}^{(1,2)} \right) - \mathbf{B}_{m\cdot}^{(1,2)} \right), \ 1 \leq m \leq n_1 n_2. \tag{10}$$

By moving the left hand side of (8) to the right, we have (8) is equivalent to

$$\left( -\mathbf{I}_{n_1-1} \ \mathbf{R}_{n_1-1}^T \right) \begin{pmatrix} \hat{\boldsymbol{\Lambda}}_{\cdot l}^1 [1 : n_1 - 1] \\ \hat{\boldsymbol{\Lambda}}_{\cdot l}^1 \left[ n_1 : \binom{n_1}{2} \right] \end{pmatrix} + \mathbf{S}_{n_2(n_1-1)\times(n_1-1)}^T \left( \hat{\boldsymbol{\Lambda}}_{\cdot l}^{(1,2)}[n_2 + 1 : n_1 n_2] \right) = 0. \tag{11}$$

Then, since $\sum_{m=1}^{n_1 n_2} \hat{\boldsymbol{\Lambda}}_{m\cdot}^{(1,2)} = 0$, checking directly that we have $\mathfrak{M} \left( \mathbf{S}_{n_1 n_2 \times n_1}^T \hat{\boldsymbol{\Lambda}}_{\cdot l}^{(1,2)} \right) = 0$ and

$$\left( \mathbf{S}_{n_1 n_2 \times n_1}^T \hat{\boldsymbol{\Lambda}}_{\cdot l}^{(1,2)} \right) [2 : n_1] = \mathbf{S}_{n_2(n_1-1)\times(n_1-1)}^T \left( \hat{\boldsymbol{\Lambda}}_{\cdot l}^{(1,2)}[n_2 + 1 : n_1 n_2] \right).$$

Since $\mathfrak{M} \left( \mathbf{R}_{n_1}^T \right) = 0$, we get $\mathfrak{M} \left( \mathbf{R}_{n_1}^T \hat{\boldsymbol{\Lambda}}_{\cdot l}^1 \right) = 0$ and checking directly that

$$\left( \mathbf{R}_{n_1}^T \hat{\boldsymbol{\Lambda}}_{\cdot l}^1 \right) [2 : n_1] = \left( -\mathbf{I}_{n_1-1} \ \mathbf{R}_{n_1-1}^T \right) \begin{pmatrix} \hat{\boldsymbol{\Lambda}}_{\cdot l}^1 [1 : n_1 - 1] \\ \hat{\boldsymbol{\Lambda}}_{\cdot l}^1 \left[ n_1 : \binom{n_1}{2} \right] \end{pmatrix} = \left( -\mathbf{I}_{n_1-1} \ \mathbf{R}_{n_1-1}^T \right) \hat{\boldsymbol{\Lambda}}_{\cdot l}^1.$$

So, we have that (11) is equivalent to

$$\mathbf{R}_{n_1}^T \hat{\boldsymbol{\Lambda}}^1_{\cdot l} + \mathbf{S}_{n_1 n_2 \times n_1}^T \hat{\boldsymbol{\Lambda}}_{\cdot l}^{(1,2)} = 0. \tag{12}$$

For (9), move right hand side to the left, we have

$$\mathbf{R}_{n_2}^T \hat{\boldsymbol{\Lambda}}_{\cdot l}^2 + \mathbf{W}_{n_1 n_2 \times n_2}^T \hat{\boldsymbol{\Lambda}}_{\cdot l}^{(1,2)} = 0. \tag{13}$$

In conclusion, we have showed that $(\hat{\mathbf{\Lambda}}_{\cdot l})^T \in \text{Row}(\mathbf{\Omega})$ is equivalent to $\hat{\mathbf{\Lambda}}_{\cdot l}$ satisfies equations (12) and (13). After that, checking directly that we have $\mathfrak{M}\left(\mathbf{S}_{n_1 n_2 \times n_1}^T \hat{\mathbf{\Lambda}}_{\cdot l}^{(1,2)}\right) = 0$ and $\mathfrak{M}\left(\mathbf{W}_{n_1 n_2 \times n_2}^T \hat{\mathbf{\Lambda}}_{\cdot l}^{(1,2)}\right) = 0$. Because of (10), for $1 \le m \le n_1$, the $m$th entry of the vector $-\left(\mathbf{S}_{n_1 n_2 \times n_1}^T \hat{\mathbf{\Lambda}}_{\cdot l}^{(1,2)}\right)$ is

$$-\frac{2}{n}\left[\frac{1}{n_1}\left(\sum_{k=1}^{n_1 n_2} \mathbf{B}_{k,l}^{(1,2)}\right) - \left(\sum_{k=1}^{n_2} \mathbf{B}_{k+n_2(m-1),l}^{(1,2)}\right)\right].$$

Also, for $1 \le m \le n_2$, we have the $m$th entry of the vector $-\left(\mathbf{W}_{n_1 n_2 \times n_2}^T \hat{\mathbf{\Lambda}}_{\cdot l}^{(1,2)}\right)$ is

$$\frac{2}{n}\left[\frac{1}{n_2}\left(\sum_{k=1}^{n_1 n_2} \mathbf{B}_{k,l}^{(1,2)}\right) - \left(\sum_{k=0}^{n_1-1} \mathbf{B}_{kn_2+m,l}^{(1,2)}\right)\right].$$

For $1 \le i \le 2$ and $1 \le j \le n_i$, since $\mathbf{A}_{j\cdot}^i = \left(\mathbf{A}_{j,1}^i, \mathbf{A}_{j,2}^i, \cdots, \mathbf{A}_{j,p}^i\right) \in \mathbb{C}^i$, we have $|\mathbf{A}_{j,k}^i| \le \mu_{ik} + \sigma_{ik}$ for $1 \le k \le p$. For $1 \le m \le n_1$, according to a direct calculation we get

$$\left|\frac{2}{n}\left[\frac{1}{n_1}\left(\sum_{k=1}^{n_1 n_2} \mathbf{B}_{k,l}^{(1,2)}\right) - \left(\sum_{k=1}^{n_2} \mathbf{B}_{k+n_2(m-1),l}^{(1,2)}\right)\right]\right|$$

$$=\left|\frac{2}{n}(n_2)\left(\frac{1}{n_1}\left(\sum_{k=1}^{n_1} \mathbf{A}_{k,l}^1\right) - \mathbf{A}_{m,l}^1\right)\right|$$

$$\le\frac{2}{n}(n_2)\left(\frac{n_1-1}{n_1}\right)(2\sigma_{1l}).$$

Similarly, for $1 \le m \le n_2$, by a direct computation we get

$$\left|\frac{2}{n}\left[\frac{1}{n_2}\left(\sum_{j=1}^{n_1 n_2} \mathbf{B}_{j,l}^{(1,2)}\right) - \left(\sum_{j=0}^{n_1-1} \mathbf{B}_{jn_1+m,l}^{(1,2)}\right)\right]\right|$$

$$=\left|\frac{2}{n}(n_1)\left(\frac{1}{n_2}\left(\sum_{k=1}^{n_2} \mathbf{A}_{k,l}^2\right) - \mathbf{A}_{m,l}^2\right)\right|$$

$$\le\frac{2}{n}(n_1)\left(\frac{n_2-1}{n_2}\right)(2\sigma_{2l}).$$

In conclusion, we have showed that

$$\|\mathbf{R}_{n_1}^T \hat{\mathbf{\Lambda}}_{\cdot l}^1\|_\infty \le \frac{2}{n}(n_2)\frac{(n_1-1)}{n_1}(2\sigma_{1l}),$$

and

$$\|\mathbf{R}_{n_2}^T \hat{\mathbf{\Lambda}}_{\cdot l}^2\|_\infty \le \frac{2}{n}(n_1)\frac{(n_2-1)}{n_2}(2\sigma_{2l}).$$

So, by Lemma 3, $\exists \hat{\mathbf{\Lambda}}_{\cdot l}^1$ satisfying (12) and $\exists \hat{\mathbf{\Lambda}}_{\cdot l}^2$ satisfying (13) s.t. the following holds

$$\|\hat{\mathbf{\Lambda}}_{\cdot l}^1\|_\infty \le \frac{2}{n}(n_2)\frac{(n_1-1)}{n_1^2}(4\sigma_{1l}), \tag{14}$$

and

$$\|\hat{\mathbf{\Lambda}}_{\cdot l}^2\|_\infty \le \frac{2}{n}(n_1)\frac{(n_2-1)}{n_2^2}(4\sigma_{2l}). \tag{15}$$

Up to now, we have constructed a $\hat{\boldsymbol{\Lambda}}$ of dimension $\binom{n}{2} \times p$ satisfying equations (12) and (13) s.t.

$$
\begin{cases}
\hat{\boldsymbol{\Lambda}}_{\cdot l}^{1} \text{ satisfies (14), for } 1 \leq l \leq p, \\
\hat{\boldsymbol{\Lambda}}_{\cdot l}^{2} \text{ satisfies (15), for } 1 \leq l \leq p, \\
\hat{\boldsymbol{\Lambda}}_{m \cdot}^{(1,2)} = \dfrac{2}{n}\left( \dfrac{1}{n_1 n_2} \left( \displaystyle\sum_{k=1}^{n_1 n_2} \mathbf{B}_{k \cdot}^{(1,2)} \right) - \mathbf{B}_{m \cdot}^{(1,2)} \right), \ 1 \leq m \leq n_1 n_2.
\end{cases}
$$

**Step 3:** Finally, we construct $\hat{\mathbf{Y}}$ and show that we can determine the cluster membership of $\mathbf{A}$ correctly if the conditions in Theorem (1) holds. Set

$$
\begin{cases}
\hat{\mathbf{Y}}_{\cdot l}^{1} = \hat{\mathbf{Y}}_{\cdot l}^{2} = 0, \ 1 \leq l \leq p, \\
\hat{\mathbf{Y}}_{m \cdot}^{(1,2)} = \left( 1 - \dfrac{n\alpha}{2\|\mathfrak{M}\left( \mathbf{B}^{(1,2)} \right)\|_2} \right) \left( \mathfrak{M}\left( \mathbf{B}^{(1,2)} \right) \right), \ 1 \leq m \leq n_1 n_2.
\end{cases}
$$

For each pair of $\mathbf{B}_{i \cdot}$ and $\boldsymbol{\Lambda}_{i \cdot}$, notice that problem (7) is equivalent to

$$
\hat{\mathbf{Y}}_{i \cdot} \in \arg \min_{\mathbf{y} \in \mathbb{R}^p} \left( \frac{1}{n}\| \left( \frac{n}{2} \boldsymbol{\Lambda}_{i \cdot} + \mathbf{B}_{i \cdot} \right) - \mathbf{y}\|_2^2 + \alpha\|\mathbf{y}\|_2 \right), \ i = 1, 2, \cdots, \binom{n}{2}. \tag{16}
$$

Then, it is easy to see that the minimizer of (16) is

$$
\begin{cases}
\left( 1 - \frac{n\alpha}{2\|\frac{n}{2}\boldsymbol{\Lambda}_{i \cdot} + \mathbf{B}_{i \cdot}\|_2} \right) \left( \frac{n}{2}\boldsymbol{\Lambda}_{i \cdot} + \mathbf{B}_{i \cdot} \right) & \text{if } \frac{2}{n}\|\frac{n}{2}\boldsymbol{\Lambda}_{i \cdot} + \mathbf{B}_{i \cdot}\|_2 > \alpha \\
0 & \text{if } \frac{2}{n}\|\frac{n}{2}\boldsymbol{\Lambda}_{i \cdot} + \mathbf{B}_{i \cdot}\|_2 \leq \alpha.
\end{cases}
$$

Then, according to the $\boldsymbol{\Lambda}$ we constructed, for $1 \leq i \leq 2$, $1 \leq l \leq \binom{n_i}{2}$ and $1 \leq h \leq n_1 n_2$, we have the following

$$
\|\frac{n}{2}\hat{\boldsymbol{\Lambda}}_{l \cdot}^{i} + \mathbf{B}_{l \cdot}^{i}\|_2 \leq w_{1,2} < d_{1,2} \leq \|\frac{n}{2}\hat{\boldsymbol{\Lambda}}_{h \cdot}^{(1,2)} + \mathbf{B}_{h \cdot}^{(1,2)}\|_2.
$$

By the construction of $\hat{\boldsymbol{\Lambda}}$ and $\hat{\mathbf{Y}}$ together with the choice of $\alpha$, conditions (12), (13), (16) are satisfied. Equivalently, conditions (7) and (6) are satisfied, so $\hat{\boldsymbol{\Lambda}}$ and $\hat{\mathbf{Y}}$ are an optimal primal and dual solution pair of (5). By the construction of $\hat{\mathbf{Y}}$, we have

$$
\begin{cases}
\hat{\mathbf{Y}}_{k \cdot}^{i} = 0, \ 1 \leq i \leq 2, 1 \leq k \leq \binom{n_i}{2}, & (17) \\
\hat{\mathbf{Y}}_{m \cdot}^{(1,2)} = \left( 1 - \dfrac{n\alpha}{2\|\mathfrak{M}\left( \mathbf{B}^{(1,2)} \right)\|_2} \right) \left( \mathfrak{M}\left( \mathbf{B}^{(1,2)} \right) \right), \ 1 \leq m \leq n_1 n_2, & (18)
\end{cases}
$$

which means $\hat{\mathbf{Y}} = \mathfrak{D}(\hat{\mathbf{X}})$, s.t.

$$
\hat{\mathbf{X}} = \begin{pmatrix} \mathbf{X}^1 \\ \mathbf{X}^2 \end{pmatrix} \text{ and } \mathbf{X}^i = \begin{pmatrix} \mathbf{X}_{1 \cdot}^{i} \\ \mathbf{X}_{2 \cdot}^{i} \\ \vdots \\ \mathbf{X}_{n_i \cdot}^{i} \end{pmatrix},
$$

where $\mathbf{X}_{j \cdot}^{i} = \left( \mathbf{X}_{j,1}^{i}, \mathbf{X}_{j,2}^{i}, \cdots, \mathbf{X}_{j,p}^{i} \right) \in \mathbb{C}^i$, $\mathbf{X}_{1 \cdot}^{i} = \mathbf{X}_{2 \cdot}^{i} = \cdots = \mathbf{X}_{n_i \cdot}^{i}$ for $i = 1, 2$ and $\mathbf{X}_{k \cdot}^{1} \neq \mathbf{X}_{l \cdot}^{2}$ for $1 \leq k \leq n_1, 1 \leq l \leq n_2$.

So, we can determine the cluster membership of $\mathbf{A}$ correctly when the conditions in Theorem 1 holds. By lemma (1), we know that $\hat{\mathbf{X}}$ is column centered. Since $\hat{\mathbf{Y}} = \mathfrak{D}(\hat{\mathbf{X}})$, by solving the following two linear equalities,

$$
\begin{cases}
\mathbf{X}_{i \cdot}^{1} - \mathbf{X}_{j \cdot}^{2} = \left( 1 - \dfrac{n\alpha}{2\|\mathfrak{M}\left( \mathbf{B}^{(1,2)} \right)\|_2} \right) \left( \mathfrak{M}\left( \mathbf{B}^{(1,2)} \right) \right) & (19) \\
n_1 \mathbf{X}_{i \cdot}^{1} + n_2 \mathbf{X}_{j \cdot}^{2} = 0, & (20)
\end{cases}
$$

we get

$$\hat{\mathbf{X}}_{i\cdot} = \begin{cases} \frac{n_2}{n_1+n_2}\left(1 - \frac{n\alpha}{2\|\mathfrak{M}(\mathfrak{D}_2(\mathbf{A}^1,\mathbf{A}^2))\|_2}\right)\mathfrak{M}\left(\mathfrak{D}_2(\mathbf{A}^1,\mathbf{A}^2)\right) & \text{if } \mathbf{A}_{i\cdot} \in \mathbb{C}^1; \\ -\frac{n_1}{n_1+n_2}\left(1 - \frac{n\alpha}{2\|\mathfrak{M}(\mathfrak{D}_2(\mathbf{A}^1,\mathbf{A}^2))\|_2}\right)\mathfrak{M}\left(\mathfrak{D}_2(\mathbf{A}^1,\mathbf{A}^2)\right) & \text{if } \mathbf{A}_{i\cdot} \in \mathbb{C}^2. \end{cases}$$

$\square$

# 6 Proof of Proposition 1

**Proposition 1. (Isometry Invariant)** *Given a data matrix* $\mathbf{A}$ *of dimension* $n \times p$ *such that each row of* $\mathbf{A}$ *is chosen from some cluster* $\mathbb{C}^i, i = 1, 2, \cdots, c$, *and* $f(\cdot)$ *an isometry of* $\mathbb{R}^p$, *we have*

$$\hat{\mathbf{X}} = \arg\min_{\mathbf{X} \in \mathbb{R}^{n \times p}} \|\mathbf{A} - \mathbf{X}\|_F^2 + \alpha \sum_{i<j} \|\mathbf{X}_{i\cdot} - \mathbf{X}_{j\cdot}\|_2$$

$$\Longleftrightarrow f(\hat{\mathbf{X}}) = \arg\min_{\mathbf{X} \in \mathbb{R}^{n \times p}} \|f(\mathbf{A}) - \mathbf{X}\|_F^2 + \alpha \sum_{i<j} \|\mathbf{X}_{i\cdot} - \mathbf{X}_{j\cdot}\|_2.$$

*This further implies that if SON successfully determines the cluster membership of* $\mathbf{A}$, *then it also successfully determines the cluster membership of* $f(\mathbf{A})$.

*Proof.* Given $\mathbf{A}$, let $\hat{\mathbf{X}}$ be the optimal solution of problem (1), i.e.

$$\hat{\mathbf{X}} = \arg\min_{\mathbf{X} \in \mathbb{R}^{n \times p}} \|\mathbf{A} - \mathbf{X}\|_F^2 + \alpha \sum_{i<j} \|\mathbf{X}_{i\cdot} - \mathbf{X}_{j\cdot}\|_2.$$

Then, $\hat{\mathbf{X}}$ reveals the cluster-membership of $\mathbf{A}$. For any $\bar{\mathbf{X}} \in \mathbb{R}^{n \times p}$, we have

$$\|f(\mathbf{A}) - \bar{\mathbf{X}}\|_F^2 + \alpha \sum_{i<j} \|\bar{\mathbf{X}}_{i\cdot} - \bar{\mathbf{X}}_{j\cdot}\|_2$$

$$= \|\mathbf{A} - f^{-1}(\bar{\mathbf{X}})\|_F^2 + \alpha \sum_{i<j} \|f^{-1}(\bar{\mathbf{X}}_{i\cdot}) - f^{-1}(\bar{\mathbf{X}}_{j\cdot})\|_2$$

$$\geq \|\mathbf{A} - \hat{\mathbf{X}}\|_F^2 + \alpha \sum_{i<j} \|\hat{\mathbf{X}}_{i\cdot} - \hat{\mathbf{X}}_{j\cdot}\|_2$$

$$= \|f(\mathbf{A}) - f(\hat{\mathbf{X}})\|_F^2 + \alpha \sum_{i<j} \|f(\hat{\mathbf{X}}_{i\cdot}) - f(\hat{\mathbf{X}}_{j\cdot})\|_2.$$

So, we have

$$f(\hat{\mathbf{X}}) = \arg\min_{\mathbf{X} \in \mathbb{R}^{n \times p}} \|f(\mathbf{A}) - \mathbf{X}\|_F^2 + \alpha \sum_{i<j} \|\mathbf{X}_{i\cdot} - \mathbf{X}_{j\cdot}\|_2.$$

Since $f$ preserves the distance between vectors, $\mathbf{A}$ and $f(\mathbf{A})$ have the same cluster-membership in the sense that if $\mathbf{A}_{i\cdot}$ and $\mathbf{A}_{j\cdot}$ are from the same cluster $\mathbb{C}^k$, then $f(\mathbf{A}_{i\cdot})$ and $f(\mathbf{A}_{j\cdot})$ are from the same cluster $f(\mathbb{C}^k)$. Because $\hat{\mathbf{X}}$ is the cluster-membership matrix of $\mathbf{A}$, $f(\hat{\mathbf{X}})$ is also the cluster-membership matrix of $\mathbf{A}$, we conclude that $f(\hat{\mathbf{X}})$ is the cluster-membership matrix of $f(\mathbf{A})$, which means we can determine the cluster-membership of $f(\mathbf{A})$ correctly. $\square$

# 7 Proof of Theorem 2

Recall that SON in the feature space can be formulated as

$$\hat{\mathbf{X}} = \arg\min_{\mathbf{X} \in \mathbb{R}^{n \times q}} \sum_{i=1}^{n} \left(\langle\phi(\mathbf{A}_{i\cdot}), \phi(\mathbf{A}_{i\cdot})\rangle - 2\langle\phi(\mathbf{A}_{i\cdot}), \mathbf{X}_{i\cdot}\rangle + \langle\mathbf{X}_{i\cdot}, \mathbf{X}_{i\cdot}\rangle\right)$$

$$+ \alpha \sum_{i<j} \sqrt{\langle\mathbf{X}_{i\cdot}, \mathbf{X}_{i\cdot}\rangle - 2\langle\mathbf{X}_{i\cdot}, \mathbf{X}_{j\cdot}\rangle + \langle\mathbf{X}_{j\cdot}, \mathbf{X}_{j\cdot}\rangle}. \tag{21}$$

**Theorem 2.** (**Representation Theorem**) *Each row of the optimal solution of Problem* (21) *can be written as a linear combination of rows of* $\mathbf{A}$, *i.e.,*

$$\hat{\mathbf{X}}_{i\cdot} = \sum_{j=1}^{n} a_{ij}\phi(\mathbf{A}_{j\cdot}).$$

*Proof.* We define the inner product on $\mathbb{R}^p$ as $\langle \mathbf{u}, \mathbf{v} \rangle = \mathbf{u}^T\mathbf{v}$ for all $\mathbf{u}, \mathbf{v} \in \mathbb{R}^p$. Then, $\mathbb{R}^p$ is a Hilbert space. Since $\mathrm{Row}(\mathbf{A})$ is a closed linear subspace of $\mathbb{R}^p$, according to the Orthogonal Decomposition theorem, we have

$$\mathbb{R}^p = \mathrm{Row}(\mathbf{A}) \oplus \mathrm{Row}(\mathbf{A})^T.$$

So, for each row $\mathbf{X}_{i\cdot}$ of $\mathbf{X} \in \mathbb{R}^{n \times p}$, we can decompose $\mathbf{X}_{i\cdot}$ into direct sum of two vectors such that one is in $\mathrm{Row}(\mathbf{A})$, the other one is in $\mathrm{Row}(\mathbf{A})^T$ i.e. $\mathbf{X}_{i\cdot} = \mathbf{u} + \mathbf{v}$ such that $\mathbf{u} \in \mathrm{Row}(\mathbf{A})$ and $\mathbf{v} \in \mathrm{Row}(\mathbf{A})^T$. Then, we can decompose any $\mathbf{X} \in \mathbb{R}^{n \times p}$ into sum of two parts $\mathbf{U}$ and $\mathbf{V}$ such that $\mathbf{X} = \mathbf{U} + \mathbf{V}$ and $\mathbf{U}_{i\cdot} \in \mathrm{Row}(\mathbf{A}), \mathbf{V}_{i\cdot} \in \mathrm{Row}(\mathbf{A})^T$ for $i = 1, 2, \cdots, n$.

We now show that the optimal solution $\hat{\mathbf{X}} \in \mathrm{Row}(\mathbf{A})$ by contraction. Suppose $\hat{\mathbf{X}} \notin \mathrm{Row}(\mathbf{A})$, then we decomose $\hat{\mathbf{X}}$ into sum of two parts $\hat{\mathbf{U}}$ and $\hat{\mathbf{V}}$ such that $\hat{\mathbf{X}} = \hat{\mathbf{U}} + \hat{\mathbf{V}}$, $\hat{\mathbf{U}}_{i\cdot} \in \mathrm{Row}(\mathbf{A})$, $\hat{\mathbf{V}}_{i\cdot} \in \mathrm{Row}(\mathbf{A})^T$ for $i = 1, 2, \cdots, n$ and exists $j$ such that $1 \leq j \leq n, \hat{\mathbf{V}}_{j\cdot} \neq 0$.

Then, we have

$$\sum_{i=1}^{n}\left( \langle \phi(\mathbf{A}_{i\cdot}), \phi(\mathbf{A}_{i\cdot})\rangle - 2\left\langle \phi(\mathbf{A}_{i\cdot}), \hat{\mathbf{X}}_{i\cdot}\right\rangle + \left\langle \hat{\mathbf{X}}_{i\cdot}, \hat{\mathbf{X}}_{i\cdot}\right\rangle \right)$$

$$+ \alpha \sum_{i<j} \sqrt{\left\langle \hat{\mathbf{X}}_{i\cdot}, \hat{\mathbf{X}}_{i\cdot}\right\rangle - 2\left\langle \hat{\mathbf{X}}_{i\cdot}, \hat{\mathbf{X}}_{j\cdot}\right\rangle + \left\langle \hat{\mathbf{X}}_{j\cdot}, \hat{\mathbf{X}}_{j\cdot}\right\rangle}$$

$$= \sum_{i=1}^{n}\left( \langle \phi(\mathbf{A}_{i\cdot}), \phi(\mathbf{A}_{i\cdot})\rangle - 2\left\langle \phi(\mathbf{A}_{i\cdot}), \hat{\mathbf{U}}_{i\cdot} + \hat{\mathbf{V}}_{i\cdot}\right\rangle + \left\langle \hat{\mathbf{U}}_{i\cdot} + \hat{\mathbf{V}}_{i\cdot}, \hat{\mathbf{U}}_{i\cdot} + \hat{\mathbf{V}}_{i\cdot}\right\rangle \right)$$

$$+ \alpha \sum_{i<j} \sqrt{\left\langle \hat{\mathbf{U}}_{i\cdot} + \hat{\mathbf{V}}_{i\cdot}, \hat{\mathbf{U}}_{i\cdot} + \hat{\mathbf{V}}_{i\cdot}\right\rangle - 2\left\langle \hat{\mathbf{U}}_{i\cdot} + \hat{\mathbf{V}}_{i\cdot}, \hat{\mathbf{U}}_{j\cdot} + \hat{\mathbf{V}}_{j\cdot}\right\rangle + \left\langle \hat{\mathbf{U}}_{j\cdot} + \hat{\mathbf{V}}_{j\cdot}, \hat{\mathbf{U}}_{j\cdot} + \hat{\mathbf{V}}_{j\cdot}\right\rangle}$$

$$= \sum_{i=1}^{n}\left( \langle \phi(\mathbf{A}_{i\cdot}), \phi(\mathbf{A}_{i\cdot})\rangle - 2\left\langle \phi(\mathbf{A}_{i\cdot}), \hat{\mathbf{U}}_{i\cdot}\right\rangle + \left\langle \hat{\mathbf{U}}_{i\cdot}, \hat{\mathbf{U}}_{i\cdot}\right\rangle + \left\langle \hat{\mathbf{V}}_{i\cdot}, \hat{\mathbf{V}}_{i\cdot}\right\rangle \right)$$

$$+ \alpha \sum_{i<j} \sqrt{\left\langle \hat{\mathbf{U}}_{i\cdot}, \hat{\mathbf{U}}_{i\cdot}\right\rangle + \left\langle \hat{\mathbf{V}}_{i\cdot}, \hat{\mathbf{V}}_{i\cdot}\right\rangle - 2\left\langle \hat{\mathbf{U}}_{i\cdot}, \hat{\mathbf{U}}_{j\cdot}\right\rangle - 2\left\langle \hat{\mathbf{V}}_{i\cdot}, \hat{\mathbf{V}}_{j\cdot}\right\rangle + \left\langle \hat{\mathbf{U}}_{j\cdot}, \hat{\mathbf{U}}_{j\cdot}\right\rangle + \left\langle \hat{\mathbf{V}}_{j\cdot}, \hat{\mathbf{V}}_{j\cdot}\right\rangle}$$

$$> \sum_{i=1}^{n}\left( \langle \phi(\mathbf{A}_{i\cdot}), \phi(\mathbf{A}_{i\cdot})\rangle - 2\left\langle \phi(\mathbf{A}_{i\cdot}), \hat{\mathbf{U}}_{i\cdot}\right\rangle + \left\langle \hat{\mathbf{U}}_{i\cdot}, \hat{\mathbf{U}}_{i\cdot}\right\rangle \right)$$

$$+ \alpha \sum_{i<j} \sqrt{\left\langle \hat{\mathbf{U}}_{i\cdot}, \hat{\mathbf{U}}_{i\cdot}\right\rangle - 2\left\langle \hat{\mathbf{U}}_{i\cdot}, \hat{\mathbf{U}}_{j\cdot}\right\rangle + \left\langle \hat{\mathbf{U}}_{j\cdot}, \hat{\mathbf{U}}_{j\cdot}\right\rangle}$$

which contradicts the optimality of $\hat{\mathbf{X}}$. Then, the lemma follows. $\square$

## References

[1] Dimitri P Bertsekas. *Convex Optimization Theory*. Universities Press. 7