[Reviews · NeurIPS 2014]

Submitted by Assigned_Reviewer_14

Summary: This paper presents a provable guarantee under what conditions the convex optimization procedure (COP) can successfully recover the correct clustering solutions. The main result is: if the samples are drawn from two cubes, each being a cluster, then COP can obtain the correct clustering solution provided the distance between two cubes is larger than a threshold value that linearly depends on the cube size and the ratio of numbers of samples in each cluster. The proof is based on the idea of "lifting", which projects the problem into a higher dimensional space that transforms the original formulation into a separable form (separating the regularization term into the sum of l_2 norm of each row). After constructing the optimal dual solution through some algebraic operations, the primal optimal solution can be obtained. Experiments on synthetic data demonstrate numerically that the distance threshold is linear to the cube size and the ratio of numbers of samples in each cluster.

Strengths:

- COP provides a convex optimization approach for clustering and as such
has the advantage of stability compared to other clustering algorithms such as K-means, Gaussian mixture models. This paper is the first work deriving theoretical guarantees providing the sufficient conditions for COP to correctly find the clustering solution.

- The proofs are technically sound. The paper is clearly written and provides synthetic experiments to validate the theoretical results.

Weaknesses:

However, the proof is only based on the setting of two clusters.

Minor Typos:
"theoretic analysis" to "theoretical analysis"
"strict convex" to "strictly convex"
Summary: This is the first work to theoretically provide the sufficient conditions under which COP can successfully recover the correct clustering solutions.

Submitted by Assigned_Reviewer_19

This paper builds a theoretical framework of the clustering problem using convex optimization procedure (COP). It presents sufficient conditions when COP for two-cluster problems will succeed. Specifically, if all samples are drawn from two cubes, and the distance between the two cubes is greater than the so called weighted size that linearly depends on the cube size and the ratio between the numbers of the samples in each cluster, COP is guaranteed to successfully recover the cluster membership. In addition, the paper also shows that the result of COP is isometry invariant. Experimental investigation validates the theoretical results.

The paper is written clearly, and the theoretical conclusion derived in this paper is interesting and significant. The sufficient conditions regarding data distributions agree with our intuition and may provide some insights for the design of a COP algorithm. The only concern is that the theoretical analysis here is designed for the specific COP formulation with just two clusters, which is not readily extendable to the scenarios with multi clusters, and doesn’t show direct insights for other convex clustering problems. Given that, the contribution of this paper is still novel and significant.
Summary: The main contribution of the paper is to build a theoretical framework of the clustering problem with two clusters using convex optimization procedure (COP). The paper is written clearly, the conclusion is interesting and the technical contribution is sound.

Submitted by Assigned_Reviewer_25

The paper revisits a convex optimization procedure for clustering and
shows that when samples are drawn from two cubes that are sufficiently
separated the method correctly recovers the cluster memberships from
the data. The paper contains a very interesting lifing technique that
is used in the proof. The paper can be improved quite a bit -- both
in terms of the quality/clarity of the writing and the presentation of the
proofs. The originality is quite high, especially in the proof of the Theorem. The technical details of the proof is quite frankly a bit too
tedious, and I think can be considerably improved. One possible such
improvement is to note that the penalty term can be written \sum_i<
j ||x_i-x_j|| = 1/2 \sum_i,j ||x_i-x_j||. This should eliminate a
bunch of complicated sum manipulations. I expect that the difference
operator can now be replaced with the further lifted vector where all
x_i-x_j are considered, so that D(E) can simply be described as
D(E)=(E kron 1 - 1 kron E) and so forth.

Also, note that the problem for p=1 is tractable and can very easily solved analytically for the two cluster case (without any lifting). Here are some other specific comments on the paper:

Authors don't cite earliest occurence of method. There is at least this earlier paper describing COP.

Pelckmans K., De Brabanter J. Suykens J.A.K., De Moor B., Convex Clustering Shrinkage, in Workshop on Statistics and Optimization of Clustering Workshop (PASCAL), London, U.K., Jul. 2005. ftp://ftp.esat.kuleuven.ac.be/pub/SISTA/ida/reports/kp05-111.pdf

It's strange to call the NP-hardness of K-means an instability. Usually an instability would refer to small perturbations leading to big changes in the result?

SON is previously used as an acronym for the method. Why invent a new acronym (COP) in this paper? That hardly seems to be properly respectful of the earlier work.

a provable guarantee when COP succeeds in identifying cluster membership. -- a provable guarantee that COP succeeds in identifying cluster membership under certain conditions?
This sentence makes it sound like you can do more than your paper does.

Isn't the operator D the same as D_1 up to a permutation? Do you really
need D?

The phrase "each cluster being a cube" is used repeatedly and is
strange. I assume you just mean that the cluster corresponds to the
points that were drawn from a uniform distribution over the cube.

Rearrange the rows of A ...
Note that such a rearrangement is not unique since the ordering of A^1
and A^2 is not prescribed

"for correctly determining" -- "to correctly determine" or "for
correct determination of"

I have to give you compliements for proving Hocking's rotation conjecture.

(3) seems overkill to state explicitly. The space could be used in a
better way.

The matrices on p.5 is presented with no intuition or clue as to what
they are to be used for. This is a case of putting the technical
messy details in the paper and the intuition in the supplemental
portion. The appendix gives the intuition for the matrix \Omega,
whereas the paper describes \Omega in gory detail with no sense of why
it's useful or where it comes from. Can you switch the appendix and
the paper with respect to the \Omega description?

What is V in equation (6). This set has not been defined.

Why are the Lemmas numbered starting with 2?

line 403: "predict how the the performance" -- "predict how the preformance"

Some additional comments for the Appendix/Supplement:

It's strange for Lemma 1 to appear in the appendix. Shouldn't this be
lemma 4 (or 3 if the others are numbered starting at 1)

page 2 appendix.

The statement after the sentence
"Then, because A is column centered, we get"
does not need the column centered property.
This is only needed on line 084-086.

line 164: column centerd -- column centered

The statement on line 188 should be shown explicitly

Based on author feedback responding to my comments I have changed my review from "marginally above the acceptance threshold" to "Good paper, accept".
Summary: The paper shows some nice results using what to me seems to be a novel
lifting technique. However, the paper suffer from being poorly
written and from proofs not having been very streamlined. This paper
is not ready for public dissemination in its present form.
Author Feedback
Author rebuttal: We thank the reviewers for their very helpful and insightful comments. We first provide responses for each main question raised by more than one of the reviewers, and then provide responses for each reviewer for the particular question raised individually.

1. The setting is based on two clusters only.
Although the setting of two clusters seems restrictive, we believe the result as well as the proof is interesting in the following points. (1) This paper is the first attempt to give a rigorous proof for why and when COP works. (2) We achieve this by lifting the data points into a higher dimensional space using the specially designed difference operator D_1 and D_2. Notice that the closed form solution in Theorem 1 depends on D_2, which provide evidence that we are taking a correct first step. (3) Perhaps most importantly, the main thrust of this paper is to explore using techniques from high-dimensional statistics, in particular regularization methods that extract low-dimensional structures such as sparsity or low rankness, to tackle *clustering problems*. We believe this work not only helps to understand why COP works in practice as shown in previous works, but also provides important insights to develop novel algorithms based on high-dimensional statistics tools for clustering tasks.

To Reviewer_14 and Reviewer_19
Thanks for the kind words and useful suggestions; we shall correct all the typos pointed out.

To Reviewer_25
1, Reviewer_25 suggests to replace the penalty term \sum_{ i< j } || x_i – x_j || by ½ \sum_i,j || x_i – x_j || and so that D(E) can be simply described as D (E) = (E kron 1 – 1 kron E).
Thanks for the suggestion and we will look into it. Also notice that when the matrix E is applied with the operator D_1 or D, the order of ( E_i – E_j ), which is the difference of ith and jth row of E, is important. Different order will lead to different matrix \Omega and different \Omega will lead to significantly different proofs, because lemma 3 depends on the structure of \Omega. So, it is not immediately clear to the authors whether simplifying the definition of D will simplify the proof.

2, Reviewer_25 suggests to add earlier paper describing COP.
Thanks a lot for the kind reminder; we will add this in the updated version.

3, Reviewer_25 mentions it is strange to call NP-hardness of k-means instability.
Actually, we say k-means is instable in the sense that Lloyd’s algorithm to tackle the k-means requires randomly initializing the clusters and different initialization may lead to significantly different final cluster results. There words are immediately after NP-hardness (at line 34, 35, 36 of paper). We are very sorry for causing the confusion.

4, Reviewer_25 asks why invent new name COP instead of SON which is used in previous work.
Thank you very much and we will change it to SON.

5, Reviewer_25 asks if the operator D and D_1 are the same up to a permutation and do we really need D?
Yes, they are same up to a permutation. But, we still need D. For example, if A is a matrix by concatenating E and F vertically. Then, the order of ( A_i – A_j ), which is the difference of ith and jth row of A, is different when applied with D and D_1. As we explained for question 1, the order of ( A_i – A_j ) is important.

6, Reviewer_25 points out that rearranging the rows of A in theorem 1 is not unique.
Yes, it is not unique, but it is unique when applied with the operator E \circ D_2, which is the composition of the average operator E and the difference operator D_2. The difference operator is defined in definition 2 and the average operator is defined in definition 3. The operator E \circ D_2 is used in theorem 1.

7, Reviewer_25 states there is no intuition or clue for the matrices on page 5.
We thank the reviewer for the comments. We will rearrange the material to explain the intuitions for these matrices in the revised version.

8, Reviewer_25 states that the set V is not defined.
Actually, the set V is defined in Step 1 of the proof sketch (on line 278 and line 279 of the paper).

9, Reviewer_25 asks why the lemma 1 appears in the appendix and the lemmas are numbered starting with 2 in main paper.
Thank you very much for pointing out this, and we will correct it in the updated version.

10, Reviewer_25 points out a typo on line 403.
Thank you very much for pointing out this, we shall correct this typo in the revised version.

11, Reviewer_25 claims that the statement after the sentence “Then, because A is column centered, we get” does not need the column centered property.
Actually, we need the column centered property, otherwise the whole part, after the sentence “Then, because A is column centered, we get”, may not equal 0.

12, Reviewer_25 points out a typo on line 164 of appendix.
Thank you very much for pointing out this typo, we will correct this.

13, Reviewer_25 suggests that the statement on line 188 of appendix should be shown explicitly.
Thank you very much for this suggestion; we will explain it in more detail.